# Targeted modulation of MMP9 and GRP78 via molecular interaction and *in silico* profiling of *Curcuma caesia* rhizome metabolites: A computational drug discovery approach for cancer therapy

Mahek Desai[1☯], Soham Bhattacharya[2☯*], Saurabhkumar Mehta[1], Kaushiki Joshi[1], Mitesh B. Solanki[3], Trilok Akhani[3], Iva Viehmannová[4], Eloy Fernández Cusimamani[4]

1 Department of Life Sciences, Parul Institute of Applied Sciences, Parul University, Waghodia, Vadodara, Gujarat, India, 2 Department of Agroecology and Crop Production, Faculty of Agrobiology, Food and Natural Resources, Czech University of Life Sciences Prague, Suchdol, Czech Republic, 3 Department of Physics, Parul Institute of Applied Sciences, Parul University, Waghodia, Vadodara, Gujarat, India, 4 Department of Crop Sciences and Agroforestry, Faculty of Tropical AgriSciences, Czech University of Life Sciences Prague, Suchdol, Czech Republic

☯ These authors contributed equally to this work.
* bhattacharya@af.czu.cz

## Abstract

Cancer remains a leading cause of mortality worldwide, with conventional therapies showing limited efficacy and high toxicity. The increasing incidence and therapeutic resistance necessitate alternative strategies. In this regard, phytochemicals have emerged as potential sources of developing safer and novel anti-cancer agents. This study employs a structure-based drug design approach, integrating molecular docking, molecular dynamics (MD) simulations, and *in silico* profiling, to investigate the anti-cancer potential of metabolites from *Curcuma caesia* rhizomes. The research targets key cancer-associated proteins, Matrix Metalloproteinase-9 (MMP9) and Glucose-Regulated Protein 78 (GRP78), identified through expression analysis, functional network mapping, and pathway enrichment as critical mediators of cancer progression and metastasis. A comprehensive molecular docking analysis of 101 bioactive compounds from *C. caesia* rhizomes identified curcumin and bis-demethoxycurcumin as promising candidates, demonstrating high binding affinities and stable interactions with MMP9 and GRP78. MD simulations further validated the stability and robustness of these interactions under dynamic physiological conditions. Pharmacological profiling, including ADMET analysis, Lipinski's rule compliance, and bioactivity scoring, revealed favorable drug-like properties for both compounds, including strong absorption, distribution, low toxicity, and potential therapeutic activities such as enzyme inhibition and nuclear receptor-mediated processes. KEGG pathway enrichment analysis confirmed their involvement in key biological pathways

**Data availability statement:** All relevant data are within the manuscript and its Supporting information files. This submission contains all raw data required to replicate the results of our study.

**Funding:** This work was financially supported by the Internal Grant Agency of the Faculty of Tropical AgriSciences, Czech University of Life Sciences Prague, IGA (Project No. 20243115 and 20243106). The funders had no role in study design, data collection and analysis, decision to publish, or preparation of the manuscript.

**Competing interests:** The authors have declared that no competing interests exist.

**Abbreviation:** ACC, Adrenocortical carcinoma; CYP, Cytochrome P450; EGFR, Epidermal growth factor receptor; EMT, epithelial-to-mesenchymal transition; ER, endoplasmic reticulum; GAG, Glycosaminoglycan; GPC3, Glypican-3; GPCRs, G protein-coupled receptors; GRP78, Glucose-Regulated Protein 78; ICM, Ion channel modulator; KEGG, Kyoto Encyclopedia of Genes and Genomes; KI, Kinase inhibitor; MD, Molecular dynamics; MMP9, Matrix Metalloproteinase-9; MRTD, Maximum Recommended Tolerated Dose; MTD, Maximum Tolerated Dose; OCT2, Organic Cation Transporter 2; Rg, Radius of gyration; RMSD, Root-mean-square deviation; RMSF, Root-mean-square fluctuation; Ro5, Lipinski's rules of five; SASA, Solvent accessible surface area; TCGA, The Cancer Genome Atlas; TKIs, Tyrosine kinase inhibitors; TPSA, Topological polar surface area; UPR, Unfolded protein response; UVM, Uveal melanoma; VDss, Volume of distribution; VEGF, endothelial growth factor.

linked to cancer progression, underscoring their therapeutic potential. The findings highlight curcumin and bis-demethoxycurcumin as promising phytochemical candidates for cancer therapy, capable of modulating MMP9 and GRP78 to suppress tumor progression. While these results provide a solid basis for their therapeutic potential, further experimental studies and clinical trials are crucial to confirm their efficacy and safety for human applications.

## Introduction

Cancer continues to be a significant global health challenge and a substantial burden on healthcare systems worldwide. In 2020 alone, an estimated 19.3 million new cancer cases were diagnosed, with approximately 10 million deaths attributed to the disease [1]. The persistent prevalence of cancer, exacerbated by ageing populations and post-pandemic challenges, highlights the urgent need for effective prevention, detection, and treatment strategies. Cancer's complex nature, including unregulated cell growth, genetic heterogeneity, and metastasis, complicates treatment development. Conventional cancer therapies, including chemotherapy, radiotherapy, and immunotherapy, are often limited by multidrug resistance, lack of target specificity, and systemic toxicity [2], necessitating the exploration of alternative therapeutic strategies. Phytochemicals, a diverse class of bioactive plant-derived compounds, have demonstrated the ability to modulate critical upstream and downstream oncogenic processes such as oxidative stress, chronic inflammation, dysregulated signaling pathways, and expression of pro-tumorigenic proteins, while exhibiting low toxicity profiles. Therefore, elucidating the mechanistic insights by which selected phytochemicals exert anti-cancer effects is essential for validating their therapeutic potential and facilitating their development as integrative agents in evidence-based cancer treatment strategies [3].

Among various bioactive plants, *Curcuma caesia*, commonly known as black turmeric, has garnered significant attention due to its potent anti-cancer, anti-inflammatory, and other medicinal properties [4]. The rhizome of *C. caesia* is rich in a variety of bioactive compounds, including curcuminoids, tannins, and flavonoids, which are integral to its pharmacological properties [5]. Moreover, research has suggested that the extract obtained from rhizomes of *C. caesia* has bioactive compounds like curcumin, bis-demethoxycurcumin, and ar-turmerone, which possess cytotoxic properties against cancer cell lines [6]. Recent studies indicate that extracts from *C. caesia* may influence signaling pathways associated with carcinogenesis, hinting at their potential for cancer-related treatments [7].

Cancer progression and metastasis are significantly influenced by proteins like Matrix Metalloproteinase-9 (MMP9) and Glucose-Regulated Protein 78 (GRP78). MMP9 facilitates extracellular matrix degradation, enabling cancer cell invasion, dissemination, and angiogenesis by attracting vascular endothelial growth factor (VEGF) [8]. Its role in migration, epithelial-to-mesenchymal transition (EMT), immune response, and tumor microenvironment formation make it a key target for anticancer

therapies. However, the complexity of its regulation and homology with other MMPs complicates the development of specific inhibitors [9]. On the other hand, GRP78, a molecular chaperone belonging to the heat shock protein localized in the endoplasmic reticulum and mitochondria. Its expression is upregulated in response to endoplasmic reticulum (ER) stress, where it plays a crucial role in maintaining protein homeostasis. When overexpressed, GRP78 contributes to key processes in cancer biology, including tumorigenesis, metastasis, angiogenesis, and resistance to therapeutic interventions [10]. Additionally, GRP78 facilitates immune evasion by modulating surface signaling proteins, thereby enabling tumor cells to circumvent immune surveillance [11]. The frequent overexpression of GRP78 in aggressive cancer phenotypes underscores its value as a prognostic biomarker and a promising therapeutic target, offering potential pathways for innovative cancer treatment strategies.

Several studies have highlighted the anti-cancer potential of *C. caesia* [7,12]. However, a comprehensive understanding of its bioactive metabolites and their mechanistic roles in cancer therapy remains largely unexplored. This study aims to address this gap by employing in-silico approaches to repurpose *C. caesia* phytochemicals as potential therapeutic agents targeting key cancer biomarkers. Specifically, it focuses on MMP9 and GRP78, which play pivotal roles in cancer progression, metastasis, and therapeutic resistance. The research investigates the functional roles of MMP9 and GRP78 in cancer development, their impact on tumour survival through protein upregulation, and the therapeutic efficacy of bioactive compounds derived from the *C. caesia* rhizome. The key methodologies employed in this study are presented as a workflow diagram (Fig 1). This diagram outlines the integration of computational techniques, including functional network analysis, pathway enrichment analysis, molecular docking studies, pharmacokinetic assessments, and KEGG pathway analysis, to ensure experimental relevance in anticancer drug development.

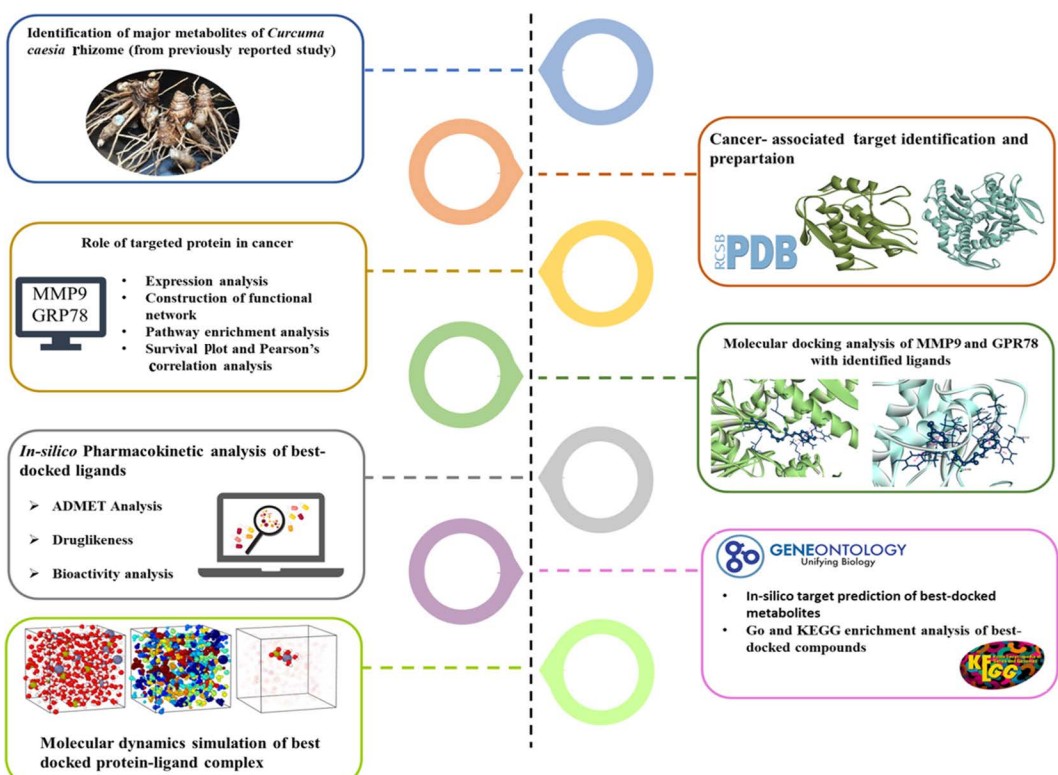

**Fig 1. Schematic diagram of the workflow of this study.**

## Materials and methods

### Gene expression profiling interactive analysis

The expression profiles of MMP9 and GRP78 across tumor samples and paired normal tissues were analyzed using the GEPIA2 server (http://gepia2.cancer-pku.cn/), with gene expression data queried from The Cancer Genome Atlas (TCGA) database. Kaplan–Meier survival plots were generated for all cancer types based on protein expression levels, using default parameters: median group cut-off and 95% confidence interval [13]. Additionally, Pearson's correlation analysis was performed to examine the relationship between the genes (MMP9, HSPA5) encoding these proteins in specific TCGA tumor datasets.

### Functional network construction and enrichment analysis

The functional and physical interactions of MMP9 and GRP78 with other proteins were analyzed using the STRING v12.0 database (https://string-db.org) by utilizing co-expressed genes, literature data, high-throughput experiments, and database mining. MMP9 and GRP78 were queried for *Homo sapiens* with a high-confidence interaction score threshold of 0.7 and a maximum of 50 interactors. The biological pathway involvement of these interactors was further examined via the Enrichr database (https://maayanlab.cloud/Enrichr/) using the KEGG 2021 Human pathway dataset. The top 10 pathways were selected based on p-value rankings for detailed analysis.

### Protein preparation

Target proteins crucial for cancer progression and metastasis were retrieved from the Protein Data Bank (PDB) (https://www.rcsb.org/). The selected targets included Matrix Metalloproteinase-9 (MMP9, PDB ID: 1GKC) and Glucose-Regulated Protein 78 (GRP78, PDB ID: 5F1X) for molecular docking analysis. Protein preparation involved removing water molecules, heteroatoms, and co-crystallized ligands using Discovery Studio Biovia. Energy minimization was performed with PyRx to optimize protein conformations for docking. The processed proteins were saved in PDBQT format to ensure compatibility with docking tools.

### Phyto-metabolites dataset preparation and ligand design

A specialized metabolite dataset of *C. caesia* rhizome was compiled through an extensive review of published articles [5,14] and phytochemical databases, particularly the PubChem server (https://pubchem.ncbi.nlm.nih.gov/). Bioactive compounds, including curcuminoids, essential oils, and polyphenols, were systematically cross-referenced with experimental data to finalize a comprehensive list of candidate metabolites (S1 Table). Chemical structures of the identified metabolites were retrieved from PubChem in SDF format with 3D representations. Ligands were prepared for docking through geometry optimization and energy minimization, and subsequently saved in PDBQT format for compatibility with molecular docking tools.

### Molecular docking analysis

Molecular docking studies were conducted using PyRx to assess the binding affinities of *C. caesia* metabolites with MMP9 and GRP78. Prepared protein and ligand structures were imported, and grid boxes were defined around the active sites of the target proteins based on prior knowledge of binding site residues and co-crystallized ligand positions. The grid dimensions were optimized to fully encompass the binding pockets. Based on the highest binding energy scores, the best binding poses were identified and visualized in Discovery Studio Biovia to analyze interactions, including hydrogen bonds, hydrophobic contacts, and π-π stacking. To ensure the reliability of the findings, docking was cross-validated using the CB-Dock2 online server (https://cadd.labshare.cn/cb-dock2/index.php, accessed on 12 October 2024). Phytochemical compounds from the metabolite dataset demonstrating the highest binding affinities to MMP9 and GRP78 were prioritized for further analysis.

### In silico pharmacokinetic analysis

Following the docking study, the top three best-docked compounds were selected for comprehensive pharmacokinetic evaluation, including ADMET analysis, Lipinski's rule of five (LRo5), bioactivity scoring, and drug-likeness assessment. Pharmacokinetic parameters related to absorption, distribution, metabolism, excretion, and toxicity (ADMET) were predicted using the pkCSM online tool (https://biosig.lab.uq.edu.au/pkcsm/prediction, accessed on October 22, 2024). Additional molecular properties of the selected compounds based on LRo5 were analyzed via SwissADME (http://www.swissadme.ch/, accessed on November 18, 2024). Bioactivity scores of the selected compounds against key human receptors, including GPCRs, ion channels, kinases, nuclear receptors, proteases, and enzymes, were evaluated using the Molinspiration web server (https://www.molinspiration.com/cgi/properties, accessed on November 18, 2024) [15].

### Component target analysis and KEGG pathway enrichment analysis

Component target analysis of the selected compounds was performed using the SwissTargetPrediction platform (http://www.swisstargetprediction.ch/, accessed on December 8, 2024), with *Homo sapiens* as the reference organism to identify relevant therapeutic targets. The predicted target data were analyzed using the Kyoto Encyclopedia of Genes and Genomes (KEGG) enrichment analysis to explore critical signaling pathways related to these compounds. This analysis was performed using the DAVID database (https://david.ncifcrf.gov/, accessed on December 27, 2024), following the methodology described by Bhattacharya et al., 2024b [16]. Visualization of the results was conducted using the Bioinformatics platform (http://www.bioinformatics.com.cn/, accessed on December 27, 2024).

### Molecular dynamics and simulation

The four most favorable protein-ligand complexes were selected for molecular dynamics (MD) simulations analysis based on their optimal binding energies and suitable docked poses identified through molecular docking studies. The macromolecular structures, represented by their PDB IDs 1GKC and 5F1X, consist of multiple well-defined subunits. These structures were prepared and visualized using the Discovery Studio Visualizer, with specific modifications applied to enhance their compatibility for in-silico virtual screening and MD simulations. A comparative analysis of the dynamic properties of the target proteins and their respective protein-ligand complexes was conducted. This was facilitated by employing the GROMACS software suite, following the protocol described by Kandasamy et al. (2022) [17]. The CHARMM36 force field was utilized to convert the protein structures from their PDB format into the GROMACS-compatible GMX format, with all required parameter files generated in accordance with guidelines from the GROMACS training materials [18].

MD simulations were performed for each protein-ligand complex over a trajectory of 100 nanoseconds (ns). Detailed analyses were conducted at 20-ns intervals to evaluate critical dynamic parameters, including solvent-accessible surface area (SASA), root-mean-square deviation (RMSD), radius of gyration (Rg), root-mean-square fluctuation (RMSF) of backbone atoms, and the number of hydrogen bonds (HB). These parameters provided insights into the conformational stability and dynamic behavior of the complexes. Graphical representations of these parameters were generated using xmgrace (https://plasma-gate.weizmann.ac.il/Grace/), offering a comprehensive visualization of the molecular dynamics results.

## Results and discussion

### Gene expression profile of MMP9 and GRP78 in TCGA pan-cancer cohort

The expression profiles of MMP9 and GRP78 across various cancer types were analyzed using data from the GEPIA2 server. Among 31 cancer types studied, both proteins were found to be upregulated in 29 cancers, while they were downregulated in three cancer types (Fig 2). MMP9 expression was notably downregulated in thymoma (THYM) and showed no significant alteration in brain lower-grade glioma (LGG) and acute myeloid leukemia (LAML), suggesting that MMP9

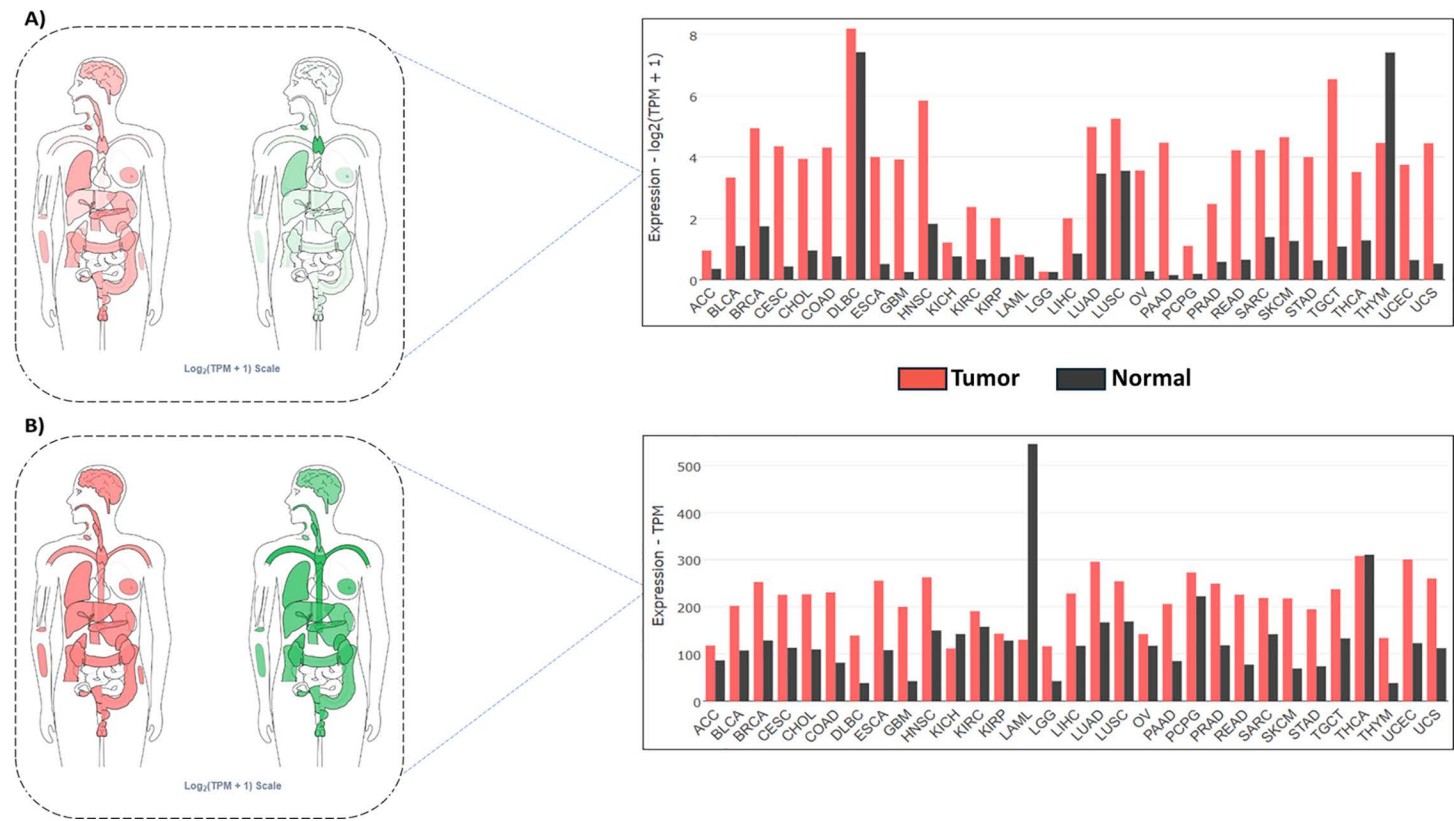

**Fig 2. Gene expression profile of A) MMP9 and B) GRP78 in pan-cancer.** All the tumor abbreviations used in this figure are mentioned in S2 Table.

does not play a significant role in these cancers. Similarly, GRP78 was downregulated in acute myeloid leukemia and kidney chromophobe (KICH) and showed no significant change in thyroid carcinoma (THCA). These findings indicate that both proteins are likely not involved in acute myeloid leukemia. However, their upregulation in most of the cancer types underscores their potential involvement in tumorigenesis.

Further analysis revealed that the overexpression of MMP9 and GRP78 correlates with reduced survival rates in cancer patients. Survival maps highlight the prognostic significance of these proteins, showing log10-transformed hazard ratios (log10(HR)) for their expression levels concerning patient survival (Fig 3A). For instance, in adrenocortical carcinoma (ACC) and bladder urothelial carcinoma (BLCA), elevated GRP78 expression, encoded by the HSPA5 gene, is strongly associated with poor survival, suggesting its oncogenic role in these cancers. Similarly, kidney renal papillary cell carcinoma (KIRP) and uveal melanoma (UVM) also exhibit significant positive correlations between GRP78 expression and worse survival outcomes, further emphasizing its potential as a prognostic marker and therapeutic target.

MMP9 showed strong positive correlations with poor survival in ACC and kidney renal clear cell carcinoma (KIRC), as well as significant associations with worse survival in UVM. These findings suggest that both MMP9 and GRP78 may share functional roles in cancer progression, particularly in ACC and UVM, where their co-regulation or interplay could contribute to tumor aggressiveness and reduced patient survival. The Pearson correlation analysis between MMP9 and HSPA5 expression levels in ACC and UVM indicates a modest but statistically significant association (R = 0.26), suggesting a possible functional linkage in processes such as stress response, tumor progression, or metastasis (Fig 3B). Despite the statistical significance, the weak magnitude of the correlation suggests that additional factors may independently

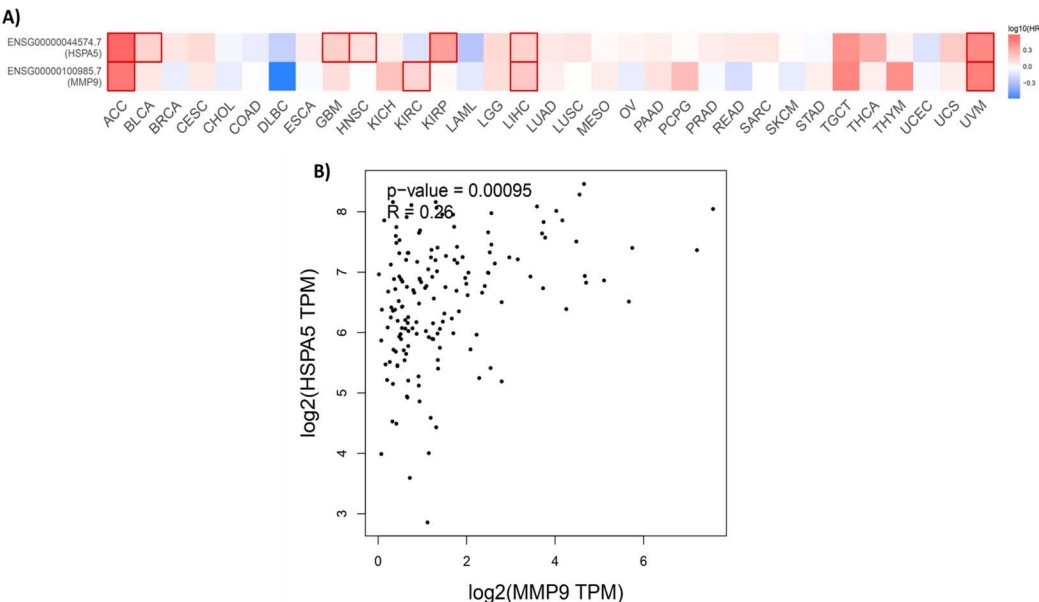

**Fig 3. A) Survival heatmap analysis of MMP9 and GRP78 (*HSPA5*) in pan-cancer.** B) Pearson correlation analysis between MMP9 and GRP78 in ACC and UVM.

regulate these proteins. This observation is supported by data from the survival heatmap (Panel A), which shows variability in protein expression patterns across different cancer types.

Functionally, GRP78 is a molecular chaperone involved in cellular stress responses and has been implicated in cancer cell survival, metastasis, and therapy resistance. Its consistent association with poor survival outcomes highlights its potential as a therapeutic target [19]. Conversely, MMP9, an enzyme involved in extracellular matrix remodeling and immune cell recruitment, exhibits context-dependent roles in survival outcomes, reflecting its variable impact on the tumor microenvironment [9]. Collectively, these findings underscore the importance of MMP9 and GRP78 in cancer biology. GRP78 demonstrates a more consistent association with poor prognosis, while MMP9's effects are cancer specific. Future research should explore their functional relationships through pathway analyses and co-expression studies in specific cancer contexts. These proteins represent promising biomarkers and therapeutic targets, offering potential avenues for novel anticancer strategies.

## Functional networking and pathway enrichment analysis of MMP9 and GRP78

To elucidate the functional relationship between MMP9 and GRP78, a functional network analysis was conducted using the STRING database. The top 50 interactors of these proteins were identified (S4 Table) and visualized in Fig 4A, C, highlighting their influence on overall cancer survival. These interactors revealed associations with various cancer-related pathways. Furthermore, the top 10 pathways involving MMP9, GRP78, and their interactors were identified based on *p*-values and adjusted *p*-values generated via the Enrichr server, as depicted in Fig 4B, D.

MMP9 interactors were found to participate in cancer-related pathways, including those involving proteoglycans, which play a critical role in cancer metastasis. Proteoglycans regulate tumor growth by interacting with growth factors through their core proteins or glycosaminoglycan (GAG) chains, as well as by GAG-independent mechanisms [20]. Glypican-3 (GPC3), a proteoglycan, has been shown to enhance cell proliferation in liver cancer via the Wnt signaling pathway [21,22]. Additionally, cell surface proteoglycans can facilitate pro-tumorigenic signaling by forming ternary complexes with

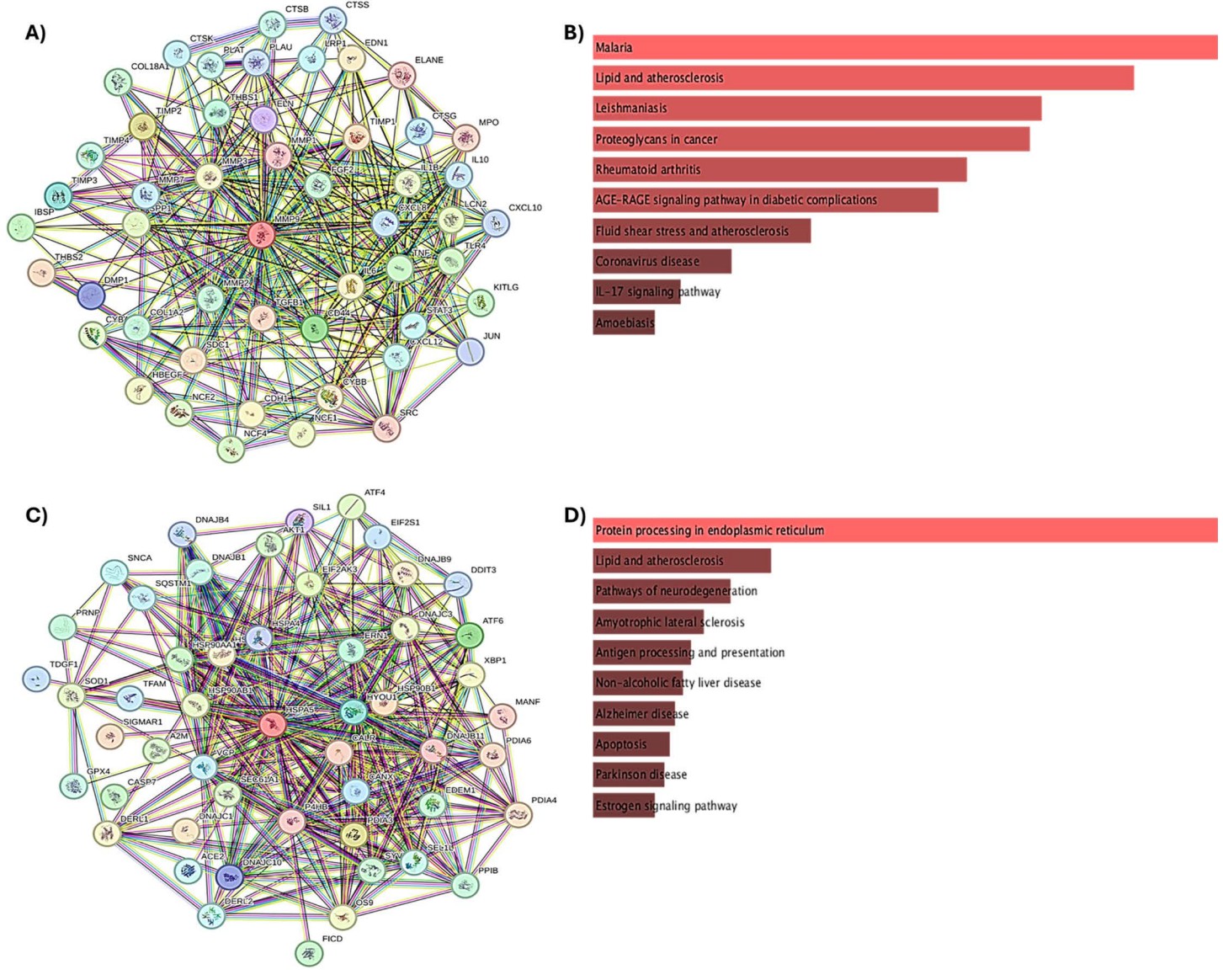

**Fig 4. Functional network and pathway enrichment analysis of MMP9 (A and B) and GRP78 (C and D).**

ligands, GAG chains, and receptors, thereby promoting growth signal self-sufficiency. This mechanism presents a promising target for cancer immunotherapy [23]. MMP9 interactors were also implicated in lipid metabolism and atherosclerosis pathways. Atherosclerosis, characterized by its inflammatory microenvironment, can significantly impact cancer progression and outcomes. Cancer and cardiovascular diseases share intricate interactions involving general and personal exposomes [24]. Furthermore, cancer cells undergo lipid metabolic reprogramming, acquiring energy and membrane resources necessary for rapid proliferation, making lipid metabolism a potential therapeutic target in cancer.

Similarly, GRP78 interactors were enriched in pathways related to protein processing in the endoplasmic reticulum (ER) and lipid metabolism. ER protein processing is critical in tumor progression, with the unfolded protein response (UPR) promoting adaptation and survival under ER stress [19]. The UPR, mediated by ER transmembrane proteins, monitors

protein folding and initiates transcriptional and translational responses to maintain homeostasis [25]. Targeting ER protein processing could serve as an alternative approach to cancer therapy. Therefore, the involvement of MMP9 and GRP78 interactors in several key cancer signaling pathways underscores their pivotal roles in cancer initiation and progression. These findings highlight MMP9 and GRP78 as promising therapeutic targets for innovative cancer treatments.

## Molecular docking and interaction analysis

Molecular docking analysis was conducted to explore the interactions between *C. caesia* rhizome-derived phytochemicals and the target proteins MMP9 and GRP78 using a dataset of 101 preidentified *C. caesia* metabolites. Binding energy calculations revealed values ranging from −3.8 kcal/mol to −8.5 kcal/mol for GRP78 and −4.5 kcal/mol to −9.1 kcal/mol for MMP9 (S3 Table). Based on binding energy criteria with both targeted proteins, curcumin and bis-demethoxycurcumin emerged as the top two candidates and were subjected to detailed interaction analysis with MMP9 and GRP78 (Table 1).

Both curcumin and bis-demethoxycurcumin exhibited a strong binding affinity towards MMP9 with a binding energy of −8 and −9.1 kcal/mol, respectively. Curcumin demonstrated significant binding interactions with MMP9 by forming two conventional hydrogen bonds. These bonds were observed at bond lengths of 2.72 Å and 1.98 Å, involving the A chain residues of GLU402 and LEU188, respectively. Additionally, curcumin established other notable interactions, including noncovalent π interactions, specifically the π-electron cloud interactions between aromatic groups arranged in a T-shaped geometry. Hydrophobic interactions, categorized as Alkyl and π-Alkyl, were identified with the amino acid residues TYR393, MET422, TYR423, ARG424, and LEU418 (Fig 5C, D). These multifaceted interactions highlight the compound's robust engagement with the MMP9 active site. Similarly, bis-demethoxycurcumin exhibited enhanced binding character-istics with MMP9, forming three strong hydrogen bonds. These interactions involved the A chain residues ALA189 (bond length: 2.72 Å), ALA191 (2.19 Å), and GLU402 (2.57 Å). Furthermore, noncovalent π interactions, including T-shaped π-electron cloud interactions, and hydrophobic interaction (Pi-Alkyl) with VAL398 were observed (Fig 5A, B). These find-ings underscore bis-demethoxycurcumin's ability to form stable and diverse interactions with the MMP9 protein.

In the protein-ligand complex with GRP78, curcumin exhibited a strong binding affinity with a calculated binding energy of −8.5 kcal/mol. The binding involved three hydrogen bond interactions with lengths of 2.57 Å, 2.51 Å, and 2.56 Å, engaging the residues ARG297, SER300, and ARG367, respectively. Additional interactions included unfavorable donor-donor interactions, amide-π stacking, and hydrophobic interactions such as Alkyl and π-Alkyl groups (Fig 6C, D).

**Table 1. Top hit interactions of MMP9 and GRP78 with ligands and their hydrogen bonds and several other interactions.**

| PDB ID | Compound Name | Binding Affinity (ΔG in kcal/mol) | No. of H-Bonds | H-Bonds and Interacting Residues | No. of other Interac-tions | Other Interactions and Numbers | Other Interaction and Interacting Residues |
|--------|---------------|------------------------------------|----------------|----------------------------------|----------------------------|--------------------------------|---------------------------------------------|
| 5F1X | Bis-demethoxycurcumin | −8.0 | 2 | ARG A:297(1), ASP A:391(1) | 8 | Unfavourable donor-donor(1), Pi-Alkyl(3), Pi-sigma(2), Pi-cation(1), amide Pi-stacked(1) | ARG A:289(3), LYS A:81(3), ILE A:61(1), ARG A:290(1) |
| | Curcumin | −8.5 | 3 | ARG A:297(1), SER A:300(1), ARG A:367(1) | 8 | Carbon-Hydrogen bond(2), Unfavourable donor-donor(1), Pi-Alkyl(2), Alkyl(2), Amide-Pi stacked(1) | ARG A:297(2), VAL A:394(1), ARG A:367(2), ASP A:224(1), LYS A:96(1), GLY A:364(1) |
| 1GKC | Bis-demethoxycurcumin | −9.1 | 3 | ALA A:189(1), ALA A:191(1), GLU A:402(1) | 6 | Pi-Pi Stacked(2), Pi-Pi T shaped(3), Pi-Alkyl(1) | PHE A:110(1), HIS A:411(1), HIS A:405(1), HIS A:401(1), VAL A:398(1), TYR A:423(1) |
| | Curcumin | −8.0 | 2 | LEU A:188(1), GLU A:402(1) | 13 | Carbon-Hydrogen bond(4), Pi-donor Hydrogen bond(1), Pi-Pi Stacked(1), Pi-Pi T shaped(1), Pi-Alkyl(4), Alkyl(1) | LEU A:418 (2), HIS A:401(1), TYR A:420(1), ALA A:417(1), HIS A:401(1), ARG A:424(2), LEU A:187(1), TYR A:393(1), MET A:422(1), TYR A:423(2) |

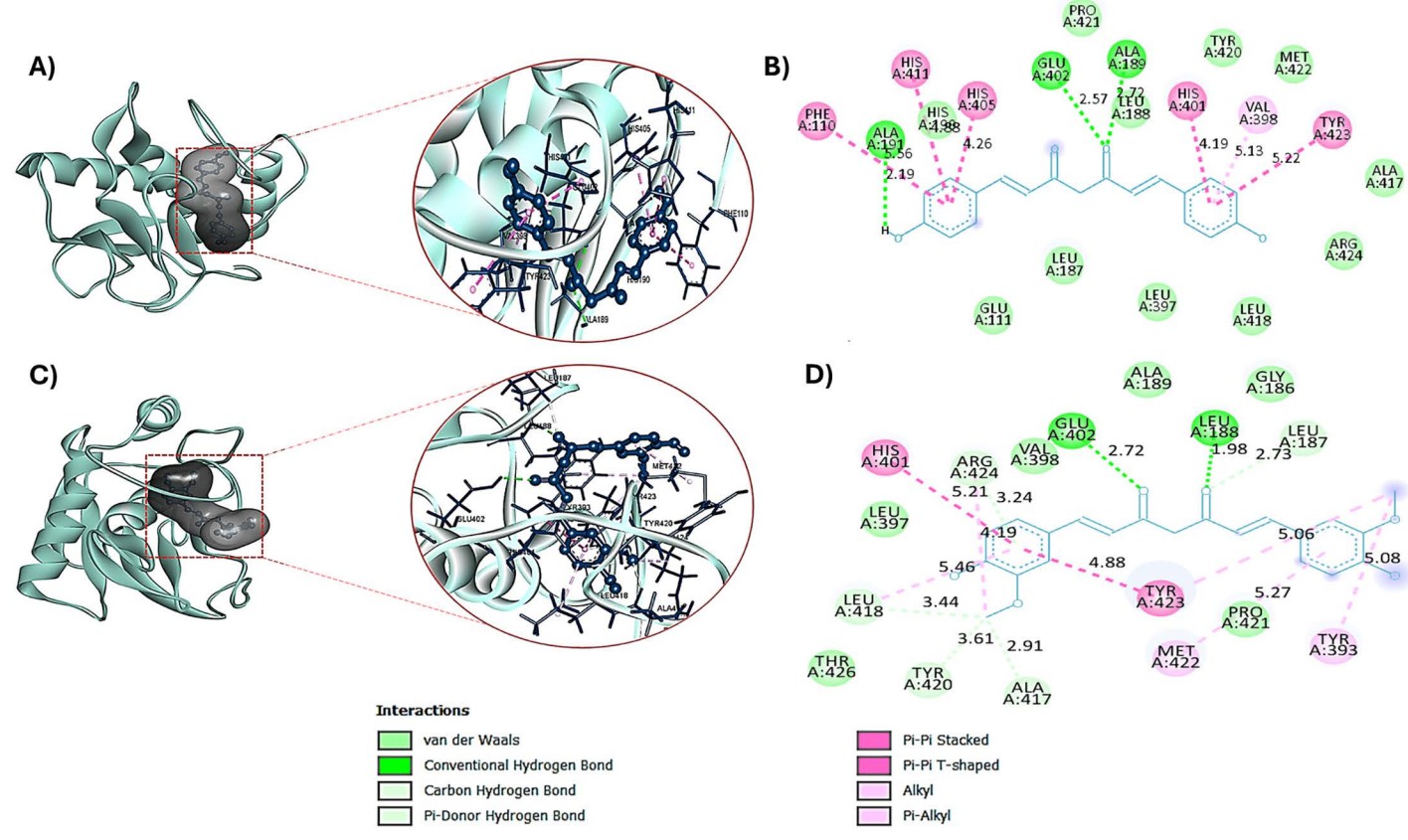

**Fig 5. Three-dimensional and two-dimensional interactions of MMP9 with bis-demethoxycurcumin (A-B) and curcumin (C-D).**

Bis-demethoxycurcumin also demonstrated a strong binding affinity towards GRP78, with a binding energy of −8 kcal/mol. This compound formed two hydrogen bonds with bond lengths of 2.69 Å and 2.07 Å, interacting with the A chain residues ARG297 and ASP391. Furthermore, non-hydrogen interactions were observed with the residues ARG289, LYS81, ILE61, and ARG290 (Fig 6A, B). These interactions further validate the compound's potential for stable and specific binding with GRP78.

Both curcumin and bis-demethoxycurcumin demonstrated high binding affinities and stable protein-ligand interactions with MMP9 and GRP78. The consistent formation of hydrogen bonds with lengths below 4 Å, coupled with diverse non-covalent and hydrophobic interactions, underscores the stability and specificity of these complexes, highlighting their potential as effective inhibitors in targeting these proteins. The strong interactions observed between two selected phyto-chemicals from *C. caesia* and the key cancer biomarkers MMP9 and GRP78 highlight their potential as effective inhibitors. Both proteins are critical in cancer progression and therapeutic resistance. Inhibition of GRP78 and MMP9 functions has been shown to impede tumorigenesis in various cancer types. However, few GRP78 inhibitors, such as PAT-SM6 and the *Escherichia coli* subtilisin cytotoxin A subunit, have been identified [26]. Also, recent years have seen growing interest in natural MMP9 inhibitors. Several such compounds, discovered via computational approaches, exhibit significant inhibitory activity, and some, like batimastat, marimastat, and prinomastat, have progressed to clinical trials [9]. Despite these advancements, there is an urgent need to develop novel, plant-derived inhibitors for both MMP9 and GRP78, aligning with contemporary therapeutic trends.

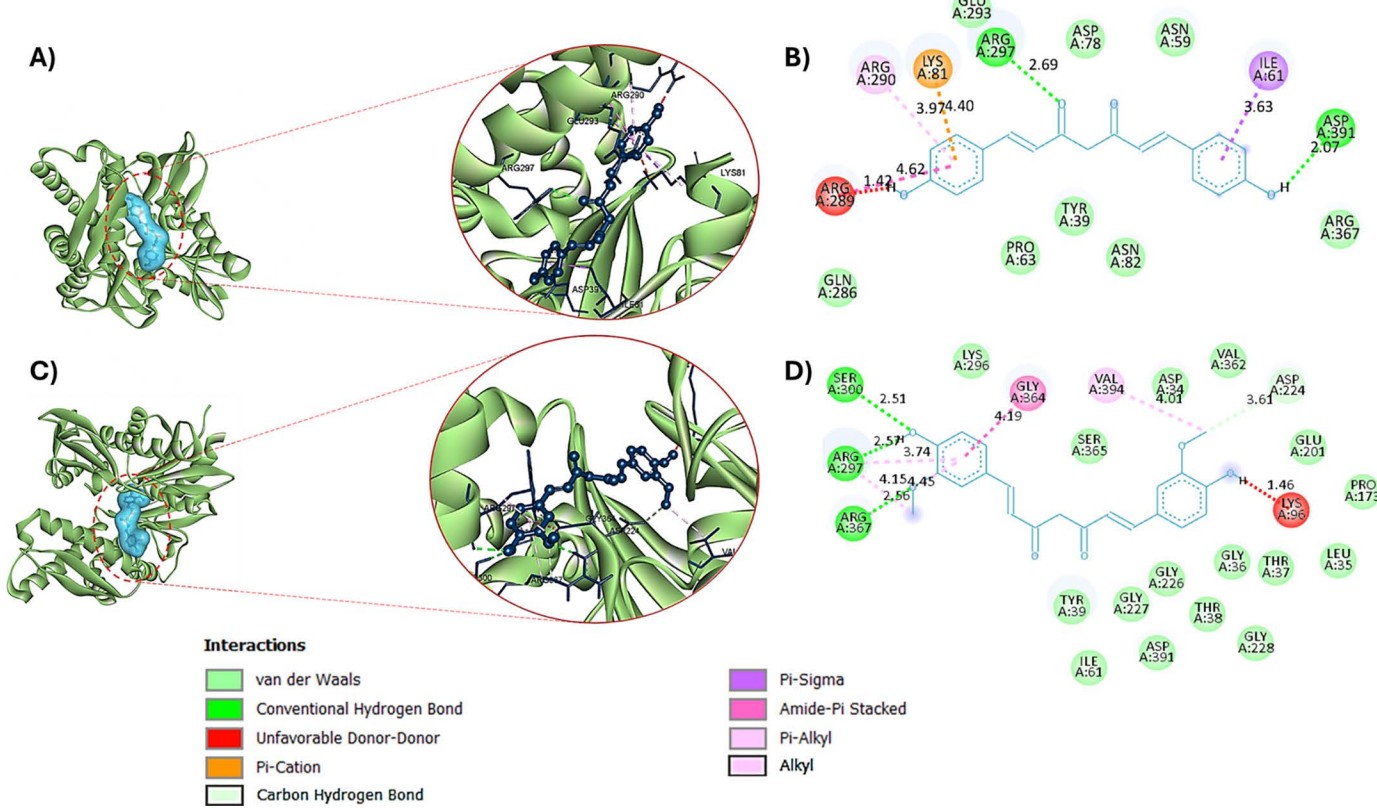

**Fig 6. Three-dimensional and two-dimensional interactions of GRP78 with bis-demethoxycurcumin (A-B) and curcumin (C-D).**

Our findings suggest that curcumin and bis-demethoxycurcumin, metabolites of *C. caesia*, effectively inhibit MMP9 and GRP78 or disrupt their interactions within cancer-related pathways. Both compounds are recognized for their potent anti-cancer properties [6,27,28]. Furthermore, prior studies have demonstrated the anti-cancer potential of *C. caesia* [7,12], yet the mechanistic roles of its bioactive metabolites remain underexplored. These findings underscore the potential of *C. caesia* phytochemicals as therapeutic agents targeting MMP9 and GRP78, offering promising avenues for cancer drug development.

## Component-target analysis and KEGG enrichment pathway analysis

To investigate and predict the molecular targets of bioactive compounds in the human body, a component-target analysis was conducted using Swiss Target Prediction. The probability threshold for target prediction was set at 0.1 (10%). The analysis identified a total of 64 potential targets for curcumin and 68 for bis-demethoxycurcumin, as detailed in S4 Table. This predictive approach serves as a foundational tool for understanding the molecular mechanisms underpinning specific phenotypes or bioactivities associated with these compounds. Additionally, it provides insights into potential off-target effects and aids in the rationalization of possible side effects [29]. The predicted targets offer valuable information for delineating the biochemical pathways influenced by curcumin and bis-demethoxycurcumin. This analysis enhances our understanding of how these compounds interact with human biological systems.

Performing KEGG enrichment analysis on the target genes associated with curcumin and bis-demethoxycurcumin, followed by bioinformatics-based visualization (Fig 7), has provided valuable insights into their molecular mechanisms

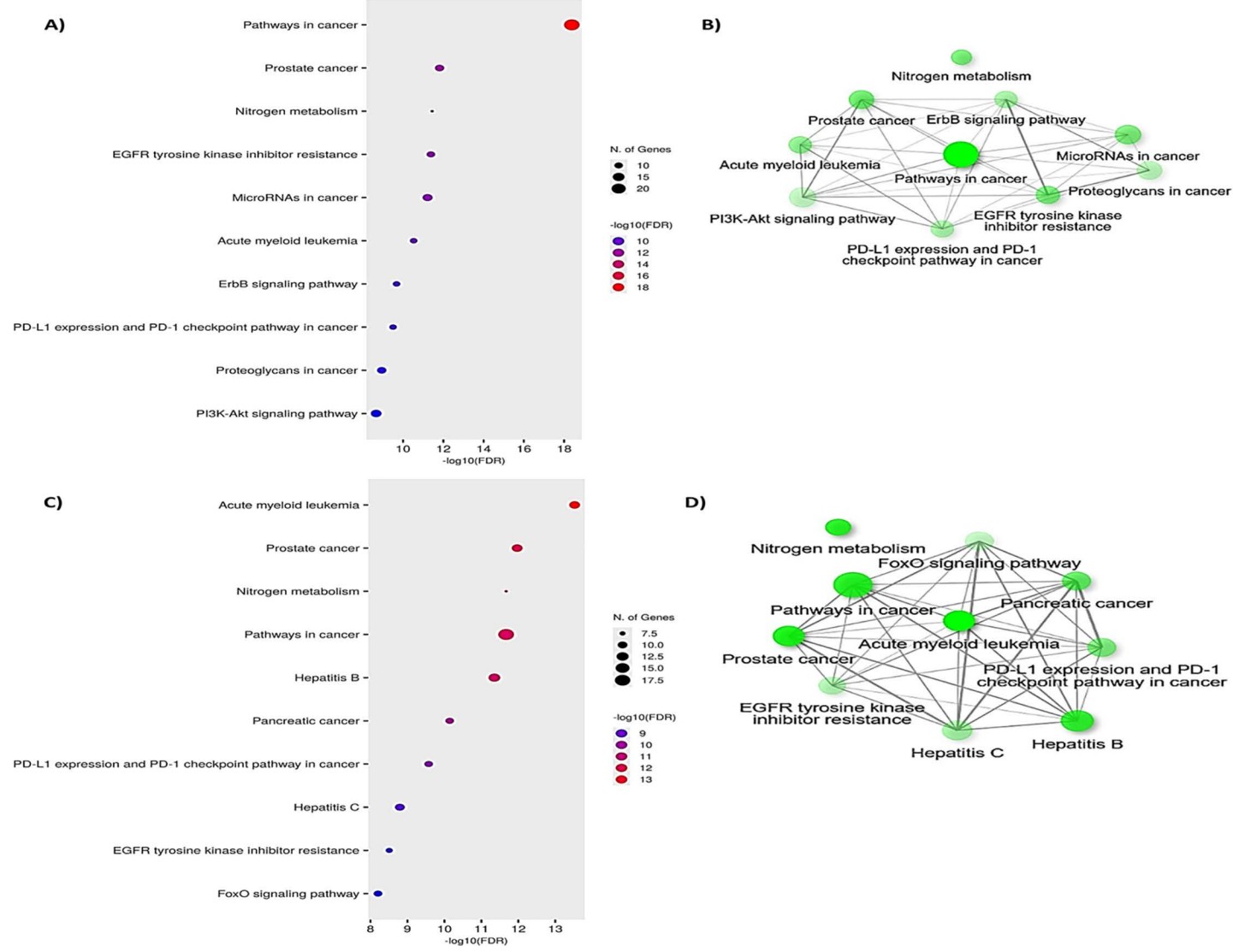

**Fig 7. KEGG analysis results for curcumin (A-B) and bis-demethoxycurcumin (C-D).**

and biological pathway involvements. The analysis identified the top 10 KEGG-enriched pathways with corrected *p*-values (h-adjusted) < 0.05, encompassing both cancer-related and physiological processes. The cancer-associated pathways identified include prostate cancer, MicroRNAs in cancer, acute myeloid leukemia, PD-L1 expression and PD-1 checkpoint in cancer, proteoglycans in cancer, pancreatic cancer, and resistance to tyrosine kinase inhibitors (TKIs) of the epidermal growth factor receptor (EGFR), as well as pathways related to ErbB receptor signaling. Additional pathways involved included the FoxO signaling pathway, nitrogen metabolism, and other disease-related pathways such as Hepatitis B and C.

These results underline the ability of curcumin and bis-demethoxycurcumin to modulate multiple cancer-associated pathways, potentially inhibiting tumor progression. For example, Mishra et al. (2024) [28] and Kaur et al. (2024) [27] have demonstrated the anti-cancer efficacy of curcumin analogs, aligning with our findings. These compounds appear to exert

their anti-cancer activity by targeting critical cancer pathways, inhibiting key molecular interactions, and offering significant therapeutic potential.

Curcumin and bis-demethoxycurcumin may act as microRNA mimics or inhibitors, modulating aberrant gene expression and restoring the balance in oncogenic or tumor-suppressor pathways. Dysregulation of ErbB receptors, including EGFR, activates signaling pathways vital for proliferation and survival, often resulting in resistance to TKIs [30]. By targeting ErbB receptors, these compounds can disrupt signaling critical for cancer cell growth, thereby circumventing resistance mechanisms. Similarly, the role of proteoglycans in the tumor microenvironment, which contributes to angiogenesis, immune evasion, and metastasis, is evident [20]. Our data also highlights the interactions of target proteins, including MMP9, in the proteoglycan cancer pathway, suggesting that these compounds can efficiently engage with MMP9-interacting proteins to inhibit tumor angiogenesis and metastasis. Moreover, these compounds may act as immune checkpoint inhibitors, similar to anti-PD-1 or anti-PD-L1 antibodies, to counter immune evasion and reawaken immune responses [31]. Additionally, evidence of effectiveness in prostate cancer, pancreatic cancer, and acute myeloid leukemia further supports their therapeutic potential. However, further studies are warranted to validate these mechanisms and evaluate their ability to specifically block these pathways, providing a promising strategy to suppress tumor growth and progression.

*In silico* **pharmacokinetic analysis.** Based on the docking study, the top-hit compounds underwent comprehensive pharmacokinetic evaluations, including **ADMET analysis**, compliance with **Lipinski's Rule of Five**, bioactivity scoring, and drug-likeness assessments, to elucidate their potential therapeutic applications and ensure human safety.

### *In silico* pharmacokinetics

#### ADMET analysis.

**Absorption** The evaluation of compound absorption predominantly relies on parameters such as skin permeability, gastrointestinal absorption, Caco-2 permeability, water solubility, and the compound's interaction with P-glycoprotein as a substrate or inhibitor. The detailed absorption data for the analyzed compounds are summarized in Table 2. The findings indicate that both compounds exhibit moderate water solubility. Notably, bis-demethoxycurcumin demonstrates a significantly higher water solubility of −3.38 log mol/L, compared to curcumin, which has a lower solubility value of −4.01 log mol/L. According to Chandra et al. (2021) [32], compounds with Caco-2 permeability values exceeding 0.90 log Papp are classified as readily permeable. The results reveal a marked disparity in Caco-2 permeability between the two compounds, with bis-demethoxycurcumin exhibiting a permeability of 0.957 log Papp, indicative of favorable intestinal barrier permeability. In contrast, curcumin shows considerably lower permeability at −0.093 log Papp. Furthermore, gastrointestinal absorption analysis, as outlined by Saha et al. (2021) [33], considers absorption rates exceeding 30% as indicative of high absorption efficiency in the human intestine, a major site for drug absorption. The results demonstrate that both compounds achieve high absorption rates within the gastrointestinal tract. Bis-demethoxycurcumin exhibits a superior absorption rate of 91.16%, compared to curcumin's absorption rate of 82.19%. The analysis further identifies both compounds as substrates of P-glycoprotein and inhibitors of P-glycoprotein-I. However, a distinct difference is observed in their interaction with P-glycoprotein-II. Bis-demethoxycurcumin does not inhibit P-glycoprotein-II, whereas curcumin demonstrates inhibitory activity. This differential inhibitory capability suggests that curcumin has the potential to bind to P-glycoprotein, a protein frequently overexpressed in human tumor tissues. Such binding may impede drug efflux, enhancing intracellular drug accumulation [34].

**Distribution.** Drug distribution is primarily assessed based on key parameters such as the human volume of distribution (VDss), and the fraction of the drug unbound in plasma. The steady-state volume of distribution (VDss) is a fundamental pharmacokinetic parameter that aids in designing appropriate drug dosage regimens. VDss represents the theoretical volume in which a drug would be distributed to achieve equivalent concentrations to those in blood plasma [35]. Higher VDss values are indicative of greater tissue distribution relative to plasma, a characteristic particularly desirable for antibiotics and antivirals aiming for extensive tissue penetration. As per established thresholds, VDss values

**Table 2. Predicted ADMET analysis.**

| Property | Model name | Compounds | |
| --- | --- | --- | --- |
| | | Bis-demethoxycurcumin | Curcumin |
| **Absorption** | Water solubility (log mol/L) | −3.38 | −4.01 |
| | Caco2 permeability (log Papp in 10−6 cm/s) | 0.957 | −0.093 |
| | GI absorption (%) | 91.159 | 82.19 |
| | P-glycoprotein substrate | Yes | Yes |
| | P-glycoprotein I inhibitor | Yes | Yes |
| | P-glycoprotein II inhibitor | No | Yes |
| **Distribution** | VDss (human) (log L/kg) | 0.139 | −0.215 |
| | Fraction unbound (human) | 0.045 | 0 |
| | BBB permeability (Log BB) | −0.089 | −0.562 |
| **Metabolism** | CYP2D6 substrate | No | No |
| | CYP3A4 substrate | Yes | Yes |
| | CYP1A2 inhibitor | Yes | Yes |
| | CYP2C19 inhibitor | Yes | Yes |
| | CYP2C9 inhibitor | Yes | Yes |
| | CYP2D6 inhibitor | No | No |
| | CYP3A4 inhibitor | Yes | Yes |
| **Excretion** | Total clearance (log ml/min/kg) | −0.008 | −0.002 |
| | Renal OCT2 substrate | No | No |
| **Toxicity** | AMES toxicity | No | No |
| | Max. tolerated dose (human) (log mg/kg/day) | −0.081 | 0.081 |
| | hERG I inhibitor | No | No |
| | hERG II inhibitor | Yes | No |
| | Oral Rat Acute Toxicity (LD50) (mol/kg) | 2.09 | 1.833 |
| | Hepatotoxicity | No | No |
| | Skin Sensitization | No | No |
| | Minnow toxicity (log mM) | 0.069 | −0.081 |

are categorized as low when log (VDss) <−0.15 and high when log (VDss) > 0.45 [35]. In the present analysis, bis-demethoxycurcumin exhibits a favorable VDss value of 0.139 log L/kg, reflecting efficient tissue distribution. In contrast, curcumin demonstrates a less favorable distribution profile with a VDss value of −0.215 log L/kg (Table 2). These results suggest that bis-demethoxycurcumin achieves superior tissue penetration compared to curcumin.

The fraction of a drug unbound in human plasma is another critical determinant of its pharmacological efficacy, as the unbound fraction is readily available for interaction with biological targets and transport across membranes. Both compounds exhibited acceptable fraction unbound values within the range of 0.02–1.0 [35]. Notably, Bis-demethoxycurcumin demonstrated a larger unbound fraction of 0.045, compared to curcumin, which exhibited an unbound fraction of 0.00 (Table 2). This highlights the increased bioavailability and potential pharmacodynamic activity of bis-demethoxycurcumin.

**Metabolism.** The cytochrome P450 (CYP) enzyme family comprises a diverse group of isozymes involved in the metabolism of a wide range of substrates, including drugs, fatty acids, steroids, bile acids, and carcinogens. The metabolic fate of a compound is significantly influenced by whether it functions as a CYP substrate or inhibitor. Our findings indicate that neither bis-demethoxycurcumin nor curcumin acts as a substrate for CYP2D6 (Table 2). However,

both compounds are substrates of CYP3A4, suggesting their metabolism is primarily mediated by cytochrome P450 3A4. This highlights a potential susceptibility to metabolic processes regulated by this isozyme. Additionally, both compounds demonstrate inhibitory activity against several key CYP enzymes, specifically CYP2C9, CYP2C19, and CYP1A2 (Table 2). This inhibitory behavior raises the possibility of drug-drug interactions, as these interactions may impede the metabolism of co-administered drugs reliant on these enzymes. Such inhibition could lead to reduced metabolic clearance, resulting in elevated plasma concentrations, increased drug accumulation, and a heightened risk of systemic toxicity [36].

**Excretion.** Excretion analysis was conducted by evaluating total clearance and determining whether GMG-ITC functions as a substrate of renal Organic Cation Transporter 2 (OCT2). OCT2 is a critical renal uptake transporter that facilitates drug accumulation and elimination via the kidneys [32]. The results indicate that neither curcumin nor bis-demethoxycurcumin acts as a substrate for OCT2 (Table 2), suggesting that their elimination occurs through alternative pathways independent of renal OCT2-mediated transport. Furthermore, both compounds exhibited total clearance values below the threshold of log (CLtot) 1.0 mL/min/kg (as detailed in Table 2), which aligns with high excretion clearance efficiency [15]. Specifically, curcumin and bis-demethoxycurcumin demonstrated total clearance values of −0.002 and −0.008, respectively (Table 2). These minimal clearance values suggest a propensity for prolonged systemic retention, which could influence the pharmacokinetic profile and therapeutic efficacy of the compounds. These findings underscore the need for further investigation into alternative excretion pathways and the potential impact of systemic retention on drug action and toxicity.

**Toxicity.** The toxicity analysis focused on key parameters, including Maximum Tolerated Dose (MTD), hERG inhibition, AMES toxicity, oral rat acute toxicity ($LD_{50}$), minnow toxicity, skin sensitization, and hepatotoxicity. AMES toxicity evaluations confirmed that both bis-demethoxycurcumin and curcumin are non-mutagenic and non-carcinogenic, providing a basis for their potential safe application in humans. The Maximum Recommended Tolerated Dose (MRTD), which estimates toxic human doses, is considered low if it falls below log 0.477 mg/kg/day [33]. The present findings reveal that the MTD for bis-demethoxycurcumin is −0.081 log mg/kg/day, while curcumin has a slightly higher MTD of 0.081 log mg/kg/day, suggesting that both compounds exhibit low toxicity profiles for human consumption (Table 2).

Additionally, neither compound exhibited hERG gene inhibition, indicating that they do not block the delayed rectifier potassium ($K^+$) channel, a property often associated with cardiac toxicity. Both compounds also showed no hepatotoxicity or skin sensitization potential, further supporting their safety profiles. In terms of oral rat acute toxicity ($LD_{50}$), both compounds demonstrated relatively high $LD_{50}$ values (refer to Table 3), signifying lower lethality compared to compounds with smaller $LD_{50}$ values [15]. For aquatic toxicity, a compound is considered acutely toxic to fathead minnows if its log $LC_{50}$ (the concentration causing 50% mortality) falls below −0.3 (i.e., less than 0.5 mM). The evaluation of minnow toxicity revealed that bis-demethoxycurcumin and curcumin exhibit significantly lower $LC_{50}$ values than this threshold, indicating relatively higher aquatic toxicity (Table 2). Consequently, their use in human applications may necessitate lower dosing to mitigate potential environmental impacts. These findings highlight the overall favorable safety profiles of both compounds, with specific considerations required for dose optimization and environmental risk assessment.

**Table 3. Drug likeness properties and predicted bioactivity score.**

| Compounds | Lipinski's Ro5 | Molecular Weight (g/mol) | H-bond acceptors | H-bond donors | Log P | TPSA (Å2) | Bioavailability score |
|---|---|---|---|---|---|---|---|
| Bis-demethoxycurcumin | Yes; 0 violation | 308.33 | 4 | 2 | 2.83 | 74.60 | 0.55 |
| Curcumin | Yes; 0 violation | 368.38 | 6 | 2 | 3.03 | 93.06 | 0.55 |

**Bioactivity score**

| Compounds | GPCRs | ICM | KI | NRL | PI | EI |
|---|---|---|---|---|---|---|
| Bis-demethoxycurcumin | 0.00 | −0.14 | −0.26 | 0.25 | −0.08 | 0.15 |
| Curcumin | −0.06 | −0.20 | −0.26 | 0.12 | −0.14 | 0.08 |

## Drug likeness and bioactivity analysis

The assessment of drug-likeness requires adherence to Lipinski's Rule of Five (Ro5), which evaluates molecular properties influencing pharmacokinetics: HBDs ≤ 5, HBAs ≤ 10, MW < 500, and LogP ≤ 5, with no more than one violation [33,37]. Both bis-demethoxycurcumin and curcumin comply with Ro5, exhibiting no violations (Table 3). Excessive lipophilicity may hinder water solubility; however, bis-demethoxycurcumin's LogP value demonstrates a favorable balance, enhancing bioavailability along with curcumin [38].

Topological polar surface area (TPSA), indicative of permeability and solubility through hydrogen bonding potential, further supports their drug-likeness. Bis-demethoxycurcumin's TPSA is 74.60 Å², reflecting favorable permeability, while curcumin's 93.06 Å² remains below the 160 Å² threshold, ensuring sufficient bioavailability [39]. Both compounds achieve a bioavailability score of 0.55, suggesting moderate bioavailability [16] (Table 3). Given their optimal molecular weights, balanced hydrogen bond donor and acceptor counts, appropriate LogP values, and acceptable TPSA, bis-demethoxycurcumin and curcumin represent promising candidates for oral drug formulations.

Bioactivity assessment for pharmaceutical substances involves parameters such as kinase and protease inhibition, enzyme activity modulation, binding affinity to G protein-coupled receptors (GPCRs) and nuclear receptors, and ion channel regulation. Molecules with bioactivity scores above 0.00 are considered biologically active, scores between −0.50 to 0.00 indicate moderate activity, and scores below −0.50 are deemed inactive [15]. The bioactivity evaluation highlights promising profiles for both bis-demethoxycurcumin and curcumin as potential drug candidates. Both exhibit strong interactions with GPCRs and nuclear receptors, significant enzyme inhibitory activity, and moderate kinase and protease inhibition capabilities (Table 3). These attributes position bis-demethoxycurcumin and curcumin as robust alternatives for therapeutic applications targeting enzyme inhibition and nuclear receptor-mediated processes, underscoring their potential in drug development.

## MD simulation analysis

Molecular dynamics (MD) simulations are invaluable for understanding protein-ligand interactions, offering detailed insights into the structural and dynamic properties of drug targets. These simulations enhance comprehension of structure-function relationships and predict how drug candidates may modulate biological targets. Additionally, MD supports structure-based drug design by accounting for the structural flexibility of drug-target systems, improving drug development precision [40]. In this study, the structural integrity, stability, and compactness of apo-proteins (ligand-free proteins) and protein-ligand complexes have been rigorously analyzed using several key parameters, including radius of gyration ($R_g$), hydrogen bonding (HB), solvent-accessible surface area (SASA), root mean square fluctuation (RMSF), and root mean square deviation (RMSD).

## RMSD analysis

The RMSD analysis of the apo forms of MMP9 and GRP78, as well as their docked complexes with bis-demethoxycurcumin and curcumin, provides detailed insights into their structural stability and ligand-induced conformational dynamics over a 100 ns MD simulation (Fig 8) [41]. The apo forms of both proteins exhibit substantial structural stability, with MMP9 demonstrating minimal backbone fluctuations within a narrow range of 0.1–0.3 nm, while GRP78 stabilizes after an initial equilibration phase, with RMSD values ranging from 0.3 to 0.6 nm.

Ligand binding results in varying degrees of structural perturbation. The bis-demethoxycurcumin complexes exhibit relatively stable RMSD ranges (0.2–0.5 nm for MMP9; 0.6–0.8 nm for GRP78), indicating moderate conformational changes upon ligand binding. In contrast, the curcumin-bound systems display higher fluctuations, reaching up to 0.6 nm for MMP9 and 1.0 nm for GRP78, suggesting that curcumin induces greater flexibility and more pronounced conformational adjustments, particularly in GRP78. Despite these ligand-induced deviations, all systems eventually achieve equilibrium,

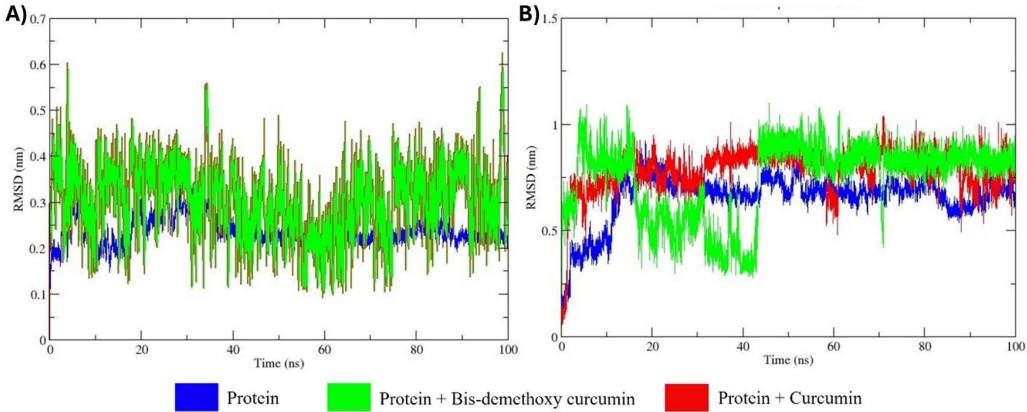

Fig 8. **Comparative assessments of the RMSD of three distinct protein backbones across a 100 ns MD run.** MMP9 is described in panel (A), while GRP78 is shown in panel (B).

maintaining overall structural integrity. The RMSD analysis also reflects the correlation between solvent-accessible surface area (SASA) and hydrophobic/hydrophilic surface interactions, highlighting ligand-induced protein folding alterations and solvation effects characterized by backbone fluctuations. These observations are further supported by the differential solvation dynamics and structural adaptability observed in ligand-bound systems [42]. These findings underscore the distinct impacts of bis-demethoxycurcumin and curcumin on the structural stability and dynamic behavior of MMP9 and GRP78. The data provides valuable insights into their binding interactions, conformational adaptability, and potential differences in binding affinities, which are critical for understanding their functional mechanisms and therapeutic applications.

### RMSF analysis

RMSF analysis of the apo forms of MMP9 and GRP78, along with their complexes with bis-demethoxycurcumin and curcumin, offers detailed insights into residual flexibility and thermodynamic stability over a 100 ns MD simulation (Fig 9) [43].

For MMP9, the apo form exhibited minimal fluctuations, with an average RMSF of ~0.18±0.01 nm, indicating high rigidity in the absence of ligand interactions. The MMP9-bis-demethoxycurcumin complex displayed moderately elevated RMSF values, particularly in residues ~900–1100 and ~1300–1500, with an average of ~0.23±0.02 nm, reflecting localized flexibility induced by stable ligand interactions [42]. In contrast, the MMP9-curcumin complex showed higher fluctuations, with peaks (~0.8 nm) around residues ~900–1100 and an average RMSF of ~0.27±0.03 nm, indicating increased conformational flexibility and weaker stabilization by curcumin. Similarly, GRP78 in its apo form exhibited high structural stability, with an average RMSF of ~0.15±0.01 nm. Upon bis-demethoxycurcumin binding, RMSF values rose moderately, particularly for residues ~1500–3000 and ~4000–5000, with an average of ~0.19±0.02 nm, suggesting localized flexibility with stable interactions. The GRP78-curcumin complex, however, displayed the highest fluctuations, with peaks (~0.3 nm) at residues ~2000–3000 and ~4000–5000, and an average RMSF of ~0.22±0.03 nm, reflecting greater flexibility and dynamic interactions.

Overall, the apo forms of MMP9 and GRP78 demonstrated the least residual flexibility, while bis-demethoxycurcumin complexes showed moderate fluctuations and stable binding. In contrast, curcumin complexes induced the highest flexibility, suggesting weaker interaction stability and significant conformational changes. These findings highlight the superior stabilizing effects of bis-demethoxycurcumin on MMP9 and GRP78 compared to curcumin, underscoring the differential binding dynamics and thermodynamic properties of these ligands [15].

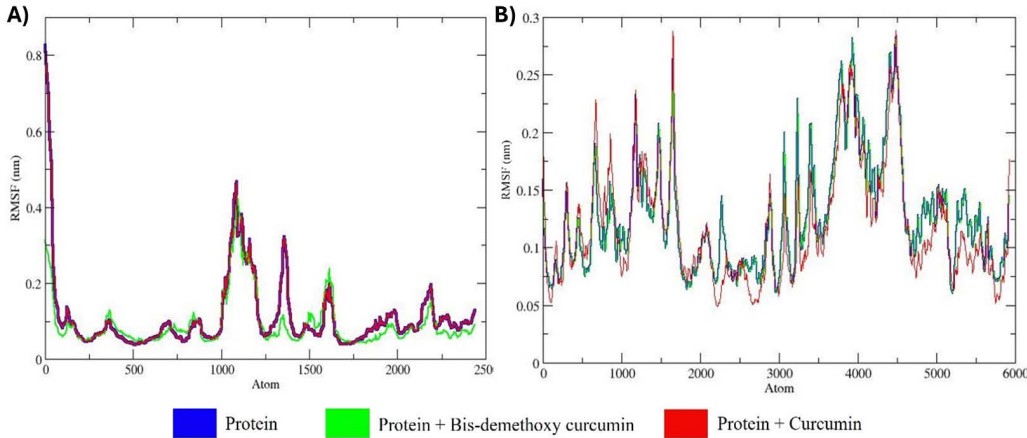

**Fig 9. Comparative RMSF profiles of successive amino acid residues in MMP9 and GRP78 during MD simulations under different conditions: apo forms (without ligands), and in the presence of bis-demethoxycurcumin or curcumin.** Panel (A) represents MMP9, while panel (B) corresponds to GRP78.

## SASA analysis

The solvent-accessible surface area (SASA) analysis of the apo forms of MMP9 and GRP78, as well as their docked complexes with bis-demethoxycurcumin and curcumin, offers critical insights into the hydrophilicity, hydrophobicity, and structural dynamics of these protein-ligand systems over a 100 ns MD simulation (Fig 10).

For MMP9, the apo form exhibited stable SASA values ranging from ~90 to ~100 nm², indicative of consistent hydrophilic behavior and solvent exposure of surface residues. The MMP9-bis-demethoxycurcumin complex showed slightly

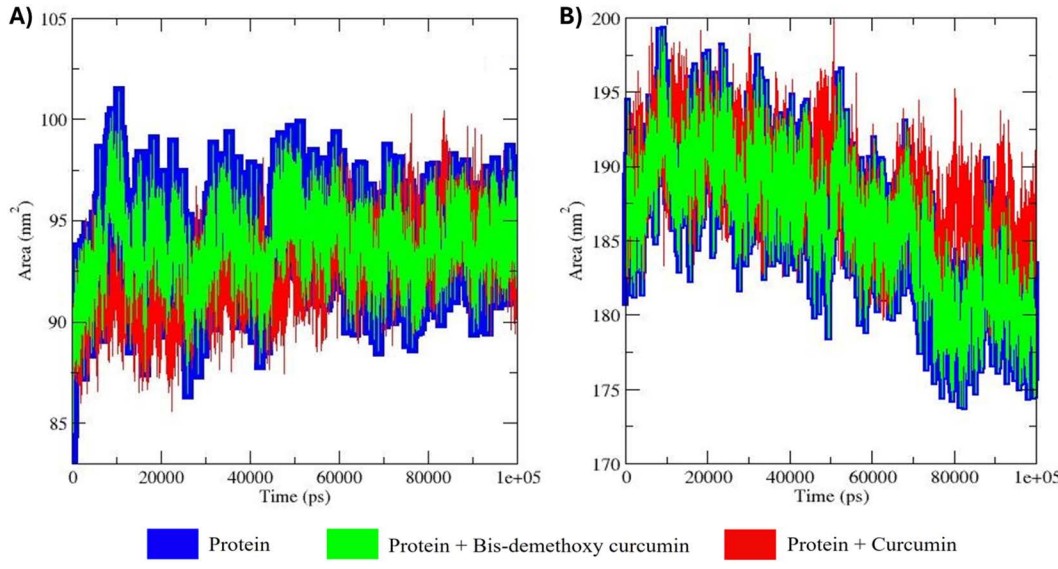

**Fig 10. SASA (Solvent Accessible Surface Area) analysis of MMP9 and GRP78 in three distinct conditions: apo form, bis-demethoxycurcumin-bound, and curcumin-bound.** The analysis is represented with specific color codes to distinguish between the environments. Panel (A) illustrates SASA variations for MMP9, while panel (B) depicts GRP78.

increased SASA values (~92 to ~102 nm²), suggesting ligand binding induced partial shielding of surface residues while preserving the protein's hydrophilic nature [44]. Conversely, the MMP9-curcumin complex demonstrated reduced SASA values (~88 to ~98 nm²), reflecting tighter ligand binding, increased hydrophobic interactions, and enhanced compactness of the protein-ligand complex [45]. Similarly, GRP78's apo form exhibited stable SASA values ranging from ~180 to ~195 nm², indicative of consistent hydrophilicity and solvent exposure. Binding of bis-demethoxycurcumin slightly decreased SASA values (~178 to ~190 nm²), indicating ligand-induced shielding and increased hydrophobicity. The GRP78-curcumin complex, however, maintained SASA values (~180 to ~195 nm²) similar to the apo form, with occasional fluctuations exceeding those of the bis-demethoxycurcumin complex. This suggests less pronounced shielding effects and greater solvent exposure, reflecting weaker structural compactness and ligand-induced stabilization [46].

Overall, while the apo forms of both MMP9 and GRP78 displayed stable SASA values, bis-demethoxycurcumin binding led to reduced solvent exposure and increased compactness. In contrast, curcumin binding was associated with relatively higher SASA values, reflecting weaker shielding and enhanced solvent exposure. These findings highlight the differential effects of bis-demethoxycurcumin and curcumin on hydrophilicity, hydrophobicity, and structural compactness, providing valuable insights into their binding interactions and stabilization of MMP9 and GRP78.

### Radius of gyration ($R_g$) analysis

The $R_g$ analysis of the apo forms of MMP9 and GRP78, as well as their docked complexes with bis-demethoxycurcumin and curcumin, provides valuable insights into their structural compactness and folding stability over a 100 ns MD simulation (Fig 11).

For MMP9, the apo form exhibited consistent $R_g$ values ranging from ~3.03 to ~3.07 nm, indicating a stable and compact structure in the absence of ligand binding. The MMP9-bis-demethoxycurcumin complex displayed nearly identical $R_g$ values (~3.04 to ~3.07 nm), suggesting that bis-demethoxycurcumin binding does not significantly alter the protein's compactness while maintaining stable interactions [47]. In contrast, the MMP9-curcumin complex showed slightly increased $R_g$ fluctuations (~3.03 to ~3.08 nm), reflecting minor destabilization and reduced structural compactness upon ligand binding.

For GRP78, the apo form maintained steady $R_g$ values between ~3.94 and ~3.97 nm, highlighting its intrinsic structural stability. The GRP78-bis-demethoxycurcumin complex exhibited similar $R_g$ values (~3.95 to ~3.98 nm) with minimal

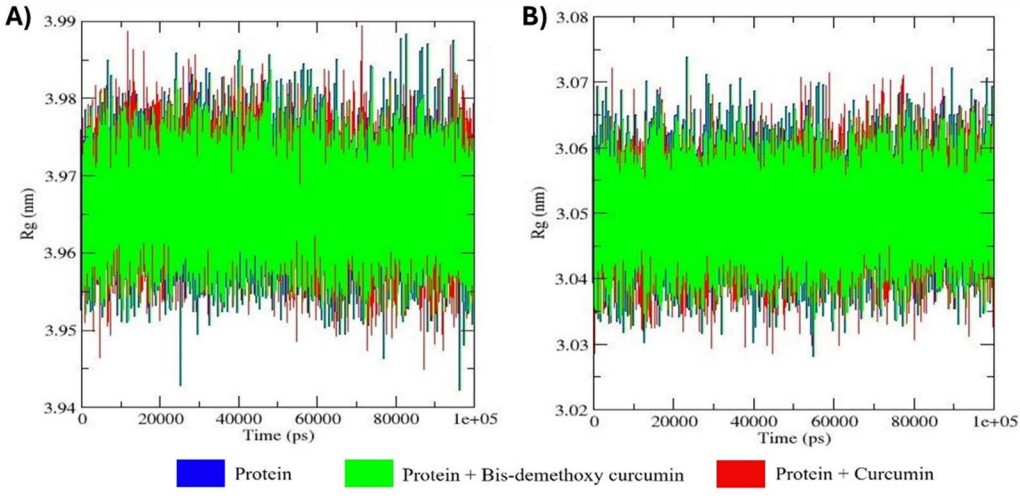

**Fig 11. Comparative analysis of the radius of gyration ($R_g$) values for different combinations, including the apo form, bis-demethoxycurcumin-bound, and curcumin-bound states.** Subfigure (A) presents results for MMP9, while subfigure (B) focuses on GRP78.

fluctuations, indicating that bis-demethoxycurcumin preserves the protein's compact folding and stability. Conversely, the GRP78-curcumin complex displayed slightly higher $R_g$ values and fluctuations (~3.94 to ~3.99 nm), suggesting reduced compactness and marginal destabilization due to curcumin binding.

Overall, the apo forms and bis-demethoxycurcumin-bound complexes of both MMP9 and GRP78 demonstrated stable and compact structures with minimal $R_g$ fluctuations. In contrast, the curcumin-bound systems exhibited slightly larger $R_g$ values and increased fluctuations, indicating minor destabilization and ligand-induced conformational adjustments. These findings emphasize the differential impacts of bis-demethoxycurcumin and curcumin on the compactness and folding integrity of MMP9 and GRP78, providing insights into their binding dynamics and structural stabilization [48].

**Analysis of hydrogen bonds based subsequent pairing analysis**

The hydrogen bond analysis of docked complexes of MMP9 and GRP78 with bis-demethoxycurcumin and curcumin, performed over a 100 ns MD simulation (Fig 12), provides detailed insights into the stability and interaction dynamics of these ligand-protein systems [15]. For GRP78, the bis-demethoxycurcumin complex consistently maintained 1–3 hydrogen bonds throughout the trajectory, with occasional peaks reaching up to 5 hydrogen bonds. This pattern reflects stable and sustained ligand-protein interactions. In contrast, the GRP78-curcumin complex displayed a more irregular hydrogen bonding profile, characterized by frequent fluctuations and intermittent bond formation, ranging between 0 and 4 hydrogen bonds, indicative of weaker and less stable interactions. For MMP9, the bis-demethoxycurcumin complex demonstrated intermittent hydrogen bond formation, with 1–3 bonds observed at various intervals, suggesting moderate binding stability. Conversely, the MMP9-curcumin complex exhibited a more consistent hydrogen bonding pattern, maintaining at least one hydrogen bond throughout the majority of the simulation and occasionally forming up to three bonds, highlighting relatively stronger and more stable interactions.

These findings align with molecular docking results and emphasize differential binding stabilities across the complexes. The sustained hydrogen bond formation in the GRP78-bis-demethoxycurcumin complex and the regular bonding pattern in the MMP9-curcumin complex underscore their respective structural contributions to stability. In contrast, the transient and irregular hydrogen bond patterns in the other complexes suggest weaker interaction dynamics [49]. This

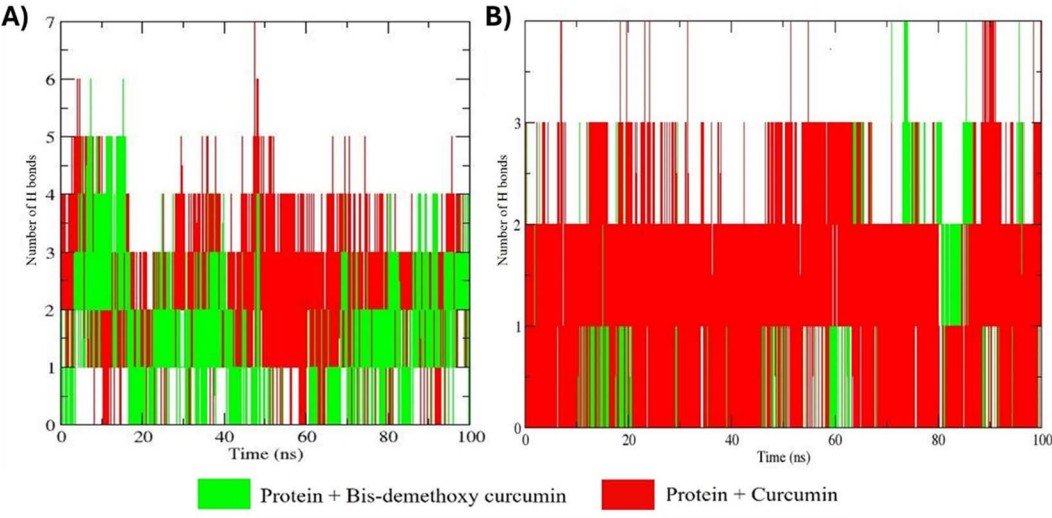

**Fig 12. Hydrogen bond intensity observed over a 100-nanosecond MD simulation.** Subfigure (A) illustrates the hydrogen bond analysis for the MMP9 protein, while subfigure (B) presents the results for the GRP78 protein.

comparative analysis provides valuable insights into the differential binding affinities and interaction mechanisms of bis-demethoxycurcumin and curcumin with MMP9 and GRP78, contributing to our understanding of their potential as therapeutic agents [50].

The MMP9 and GRP78 systems demonstrate pronounced solvation effects, accompanied by stable hydrogen bonds and consistent binding dynamics. MD simulations corroborate the docking results, confirming the stability of the complexes formed between these proteins and bis-demethoxycurcumin or curcumin. The interaction dynamics are characterized by high structural compactness and minimal fluctuations, underscoring the robust binding properties of these ligand-protein systems. However, A critical consideration in targeting GRP78 is its differential role in cancer versus normal cells. In tumors, GRP78 is often overexpressed and relocalized to the cell surface, where it drives survival, metastasis, and therapy resistance under ER stress and hypoxia. In contrast, normal cells restrict GRP78 to the ER, making surface GRP78 a viable and selective target. To ensure this selectivity, it is crucial to target surface GRP78. There are several examples of surface-specific selective binding of natural ligands with GRP78. For instance, natural compounds such as epigallocatechin gallate inhibit the ATPase domain of GRP78, while hydroxytyrosol and Caffeic acid phenethyl ester bind the SBDβ domain, demonstrating selective in silico affinity for tumor-expressed GRP78 [51,52]. Also, there are several other ways to maintain this selectivity, such as targeting extracellular GRP78 with peptides like Pep42 or monoclonal antibodies avoid off-target effects on ER-localized GRP78 in healthy cells [53]. Additionally, GRP78-targeting nanoparticles offer tumor-specific drug delivery [54], while GRP78 expression profiling via immunohistochemistry or imaging supports patient stratification [55]. Combining these approaches with phytochemicals and conventional chemotherapies further enhances tumor-specific ER stress and apoptotic responses [56].

## Conclusion

The present study investigated the roles of two critical proteins, MMP9 and GRP78, in cancer progression and evaluated their potential as therapeutic targets using a multifaceted *in silico* approach. This included expression analysis, functional networking, and pathway enrichment studies. Additionally, the study assessed the therapeutic potential of *C. caesia* rhizome-derived metabolites in modulating these target proteins through structure-based drug design, incorporating molecular docking and MD simulations.

Results identified bis-demethoxycurcumin and curcumin as potent bioactive metabolites from *C. caesia* rhizome, exhibiting favorable docking scores with MMP9 and GRP78. These compounds demonstrated strong interactions with critical amino acid residues within the proteins' active sites, effectively inhibiting their functional activity. MD simulations validated the docking results, confirming the stability, compactness, and integrity of the protein-ligand complexes over the simulation period. Further, KEGG pathway enrichment analysis revealed that bis-demethoxycurcumin and curcumin are able to modulate multiple cancer-associated pathways, highlighting their potential application for cancer therapy. Moreover, Pharmacokinetic profiling and bioactivity predictions supported the drug-like characteristics of bis-demethoxycurcumin and curcumin, suggesting their suitability as candidates for drug development. This systematic investigation lays the groundwork for subsequent in vitro and in vivo studies to elucidate the molecular mechanisms underlying the modulation of MMP9 and GRP78. Furthermore, these findings emphasize the potential of bis-demethoxycurcumin and curcumin to counteract tumorigenesis through multi-targeted interventions. Phytochemical-based drug discovery using a computational approach provides an efficient, cost-effective strategy for novel anticancer drug discovery. This *in silico* approach enables high-throughput screening and detailed molecular interaction profiling, offering a valuable alternative to labor-intensive laboratory methods. However, its predictive accuracy is limited by dependency on structural data and the inability to fully capture pharmacokinetics, toxicity, protein dynamics, and off-target effects. To improve reliability, integration of deep learning models and advanced computational tools is essential. Furthermore, comprehensive preclinical validation, including cytotoxicity assays, target and downstream protein regulation via western blotting or qRT-PCR), cancer biomarker analysis, and in vivo studies, is crucial to elucidate the mechanistic roles of candidate phytochemicals in targeted cancer therapies.

## Supporting information

**S1 Table. Phytochemical dataset of *C. caesia* rhizome.**
(PDF)

**S2 Table. Tumour abbreviations list.**
(PDF)

**S3 Table. Binding free-energy values (ΔG in kcal/mol) and interactions of selected *C. caesia*rhizome metabolites with GRP78 and MMP9.**
(PDF)

**S4 Table. Molecular targets of curcumin and bis-demethoxycurcumin predicted from.**
(PDF)

## Acknowledgments

The authors also acknowledged Mr. Sahilkumar Radadiya for his consultation and technical assistance in the MD simulation work and his support during the research process.

## Author contributions

**Conceptualization:** Soham Bhattacharya.

**Data curation:** Mahek Desai, Soham Bhattacharya, Kaushiki Joshi.

**Formal analysis:** Mahek Desai, Soham Bhattacharya, Saurabhkumar Mehta, Kaushiki Joshi, Mitesh B. Solanki, Trilok Akhani, Iva Viehmannová, Eloy Fernández Cusimamani.

**Funding acquisition:** Iva Viehmannová, Eloy Fernández Cusimamani.

**Investigation:** Mahek Desai, Soham Bhattacharya, Kaushiki Joshi.

**Methodology:** Mahek Desai, Soham Bhattacharya.

**Software:** Soham Bhattacharya.

**Supervision:** Saurabhkumar Mehta, Mitesh B. Solanki.

**Visualization:** Mahek Desai, Soham Bhattacharya.

**Writing – original draft:** Mahek Desai, Soham Bhattacharya.

**Writing – review & editing:** Saurabhkumar Mehta, Kaushiki Joshi, Mitesh B. Solanki, Trilok Akhani, Iva Viehmannová, Eloy Fernández Cusimamani.

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
