## [Decision Letter · Decision Letter 0]

PONE-D-25-13435Targeted modulation of MMP9 and GRP78 via molecular interaction and in silico profiling of Curcuma caesia rhizome metabolites: A computational drug discovery approach for cancer therapyPLOS ONE

Dear Dr. Bhattacharya,

Thank you for submitting your manuscript to PLOS ONE. After careful consideration, we feel that it has merit but does not fully meet PLOS ONE’s publication criteria as it currently stands. Therefore, we invite you to submit a revised version of the manuscript that addresses the points raised during the review process.

We look forward to receiving your revised manuscript.

Kind regards,

Prakash Palaniswamy, Ph.D

Academic Editor

PLOS ONE

Journal Requirements:

“This work was financially supported by the Internal Grant Agency of the Faculty of Tropical AgriSciences, Czech University of Life Sciences Prague, IGA (Project No. 20243115 and 20243106).”

4. We note that your Data Availability Statement is currently as follows: All relevant data are within the manuscript and in Supporting Information files.

5. Please ensure that you refer to Figures 8-12 in your text as, if accepted, production will need this reference to link the reader to the figure.

Additional Editor Comments:

Evaluate the strengths and limitations of relying solely on in silico methods for drug discovery.

What additional studies would you recommend to validate the therapeutic potential of Curcuma caesia metabolites?

If curcuzederone shows promise in computational models, what preclinical steps are necessary before clinical trials?

Discuss the potential impact of targeting GRP78 on normal cells versus cancer cells. How can this selectivity be ensured?

Reviewers' comments:

Reviewer's Responses to Questions

**Comments to the Author**

1. Is the manuscript technically sound, and do the data support the conclusions?

Reviewer #1: Yes

2. Has the statistical analysis been performed appropriately and rigorously? 

Reviewer #1: Yes

3. Have the authors made all data underlying the findings in their manuscript fully available?

Reviewer #1: Yes

4. Is the manuscript presented in an intelligible fashion and written in standard English?

Reviewer #1: Yes

5. Review Comments to the Author

Reviewer #1: Manuscript Number: PONE-D-25-13435

Abstract: This part is well furnished that details the outcome and findings. Yet the preceding part seems to be vague and better to keep it crispier with 2lines.

Introduction: Seem to be very clear about the workplan. The author should concentrate on pinpointing the importance of the study rather than explaining its importance in with vast forum, especially the first paragraph.

A complete analysis of the isolated constituents has been well explicated with detailed analysis. A distinct analysis with appropriate parameters highlights the piece of work. All the figures and tables pertaining to the methodologies are appreciable which supports the reader’s understanding.

I hereby, give my consent for the acceptance of the paper, after trimming the abstract and introduction contents.

6. PLOS authors have the option to publish the peer review history of their article (what does this mean? ). If published, this will include your full peer review and any attached files.

**Do you want your identity to be public for this peer review?** For information about this choice, including consent withdrawal, please see our Privacy Policy .

Reviewer #1: **Yes: ** Dr.S.Deepika Priyadharshini

---

## [Author Response · Author response to Decision Letter 1]

20 Jun 2025

Response to Additional Editor’s Comments:

Comment 1: Evaluate the strengths and limitations of relying solely on in silico methods for drug discovery.

Response: We thank the additional editor for their time and effort in reviewing our manuscript and for providing this insightful suggestion to improve the clarity and depth of our study.

We acknowledge that relying solely on in silico methods for phytochemical-based drug discovery, such as molecular docking, functional network mapping, pathway enrichment analysis, and ADMET predictions, offers significant advantages. These include high-throughput screening capabilities, cost-effectiveness, and the ability to generate detailed insights into molecular interactions. These tools are particularly valuable for prioritizing candidate compounds and narrowing the scope for in vitro and in vivo testing.

However, we also recognize the limitations inherent in computational approaches. In silico methods heavily depend on available structural data and may not accurately reflect complex biological processes, such as protein flexibility, cellular context, pharmacokinetics, toxicity, and off-target interactions. As such, these predictions, while useful, must be validated through experimental studies.

To address this important point, we have included a short paragraph in the Conclusion section of the revised manuscript to reflect the strengths and limitations of in silico methodologies and to guide future research. The newly added lines are as follows:

Line 753-763: “Phytochemical-based drug discovery using a computational approach provides an efficient, cost-effective strategy for novel anticancer drug discovery. This in silico approach enables high-throughput screening and detailed molecular interaction profiling, offering a valuable alternative to labor-intensive laboratory methods. However, its predictive accuracy is limited by dependency on structural data and the inability to fully capture pharmacokinetics, toxicity, protein dynamics, and off-target effects. To improve reliability, integration of deep learning models and advanced computational tools is essential. Furthermore, comprehensive preclinical validation, including cytotoxicity assays, target and downstream protein regulation via western blotting or qRT-PCR), cancer biomarker analysis, and in vivo studies, is crucial to elucidate the mechanistic roles of candidate phytochemicals in targeted cancer therapies.”

Comment 2: What additional studies would you recommend to validate the therapeutic potential of Curcuma caesia metabolites?

Response: We sincerely thank the editor for highlighting this important point regarding the need for further validation of the therapeutic potential of C. caesia metabolites. While our in silico findings provide a strong preliminary basis for their potential anticancer activity, comprehensive experimental validation is essential. Accordingly, we propose the following multi-tiered approach to systematically validate the bioactivity and therapeutic relevance of the selected metabolites:

1. In Vitro Cytotoxicity and Mechanistic Studies

• Dose-dependent cytotoxicity assays:

Initial screening should involve MTT or resazurin assays to determine LC₅₀ values in various cancer cell lines. This will establish the concentration range in which the metabolites are effective without nonspecific toxicity.

• Selection of target-relevant cell lines:

Based on the expression profile of the target proteins (e.g., GRP78, MMP9), suitable cancer cell lines should be selected using databases such as the Human Protein Atlas to ensure biological relevance.

• Mechanism of action (MoA) studies:

a) Gene/Protein Expression Analysis: Quantitative RT-PCR and Western blotting should be employed to assess the expression of target proteins and downstream signaling molecules.

b) Apoptosis Assays: Use Annexin V/PI staining and flow cytometry to quantify apoptosis, along with caspase-3/7 activity assays and analysis of apoptotic markers such as cleaved PARP and Bax/Bcl-2 ratios.

c) Live/Dead Cell Assays: Calcein-AM and propidium iodide dual staining can provide a real-time assessment of cell viability and death.

d) Lipid Droplet Assays: Oil Red O staining or BODIPY-based lipid imaging may be used to evaluate changes in lipid metabolism, a potential mechanism of anticancer activity.

2. Omics-Based Mechanistic Elucidation

• Transcriptomic and Proteomic Profiling: High-throughput RNA sequencing or proteomics can be utilized to explore the global impact of metabolites on signaling pathways such as PI3K/AKT, NF-κB, and particularly GRP78-regulated stress responses.

3. In Vivo Preclinical Validation

• Xenograft Tumor Models: Testing in suitable mouse xenograft models will help validate tumor growth inhibition, modulation of target proteins (GRP78, MMP9), and systemic toxicity. This step is critical to assess the translational potential of the compounds.

Based on these proposed future validation studies, we have included a concise paragraph in the Conclusion section that highlights the essential in vitro and in vivo experimental steps required to validate the anticancer potential of Curcuma caesia metabolites. The added lines are as follows:

Line 753-763: “Phytochemical-based drug discovery using a computational approach provides an efficient, cost-effective strategy for novel anticancer drug discovery. This in silico approach enables high-throughput screening and detailed molecular interaction profiling, offering a valuable alternative to labor-intensive laboratory methods. However, its predictive accuracy is limited by dependency on structural data and the inability to fully capture pharmacokinetics, toxicity, protein dynamics, and off-target effects. To improve reliability, integration of deep learning models and advanced computational tools is essential. Furthermore, comprehensive preclinical validation, including cytotoxicity assays, target and downstream protein regulation via western blotting or qRT-PCR), cancer biomarker analysis, and in vivo studies, is crucial to elucidate the mechanistic roles of candidate phytochemicals in targeted cancer therapies.”

Comment 3: If curcuzederone shows promise in computational models, what preclinical steps are necessary before clinical trials?

Response: We sincerely thank the editor for this insightful comment. If curcuzederone, a Curcuma caesia-derived metabolite, demonstrates promising results in computational models, several critical preclinical steps must be undertaken prior to advancing into clinical trials. These steps can be broadly categorized as follows:

1. Mechanism of Action (MoA) Validation: As outlined in our earlier response, it is essential to experimentally confirm the anticancer effects of curcuzederone through a series of in vitro studies, such as cytotoxicity assays, target protein expression analysis, and apoptosis pathway investigation, followed by in vivo validation in relevant animal models.

2. In Vitro ADME-T Profiling:A comprehensive evaluation of absorption, distribution, metabolism, excretion, and toxicity (ADME-T) properties is essential to determine the compound's pharmacokinetic behavior. Parameters such as metabolic stability, membrane permeability, and cytotoxicity in liver and kidney cells help assess drug-likeness and filter out candidates with poor bioavailability or toxicity.

3. Formulation Optimization: Given that many phytochemicals, including curcuminoids, exhibit limited solubility and poor bioavailability, formulation strategies such as nanoparticle conjugation, liposomes, or polymer-based drug delivery systems should be explored to improve systemic delivery and therapeutic efficacy.

4. Safety Pharmacology: It is crucial to evaluate potential off-target effects on major physiological systems (e.g., cardiovascular, central nervous, and respiratory systems). These studies must be conducted in accordance with Good Laboratory Practice (GLP) and align with regulatory frameworks such as ICH and FDA guidelines to support an eventual Investigational New Drug (IND) application.

5. In Vivo Efficacy and Toxicity Studies: Animal models should be used to assess pharmacokinetics, pharmacodynamics, and therapeutic efficacy. Additionally, acute and chronic toxicity studies—including LD₅₀ determination, organ histopathology, and hematological profiling—are necessary to ensure safety and establish dosing regimens.

To address this point along with our primary focus on in silico study, in the revised manuscript, we have incorporated a concise summary of these recommended preclinical steps in the Conclusion section (Lines 753–763), which provides a forward-looking perspective on the translational potential of curcuzederone and guides future experimental directions.

Comment 4: Discuss the potential impact of targeting GRP78 on normal cells versus cancer cells. How can this selectivity be ensured?

Response: We sincerely thank the editor for highlighting this important and impactful aspect regarding the therapeutic selectivity of targeting GRP78. In response to this valuable suggestion, we have included a dedicated paragraph in the Discussion section that addresses the differential expression of GRP78 in normal versus cancer cells and outlines potential strategies to enhance selectivity. We believe that this addition strengthens the scientific rigor and credibility of the manuscript while improving its clarity and overall readability. The newly added lines are as follows:

Line 717-732: “However, A critical consideration in targeting GRP78 is its differential role in cancer versus normal cells. In tumors, GRP78 is often overexpressed and relocalized to the cell surface, where it drives survival, metastasis, and therapy resistance under ER stress and hypoxia. In contrast, normal cells restrict GRP78 to the ER, making surface GRP78 a viable and selective target. To ensure this selectivity, it is crucial to target surface GRP78. There are several examples of surface-specific selective binding of natural ligands with GRP78. For instance, natural compounds such as epigallocatechin gallate inhibit the ATPase domain of GRP78, while hydroxytyrosol and Caffeic acid phenethyl ester bind the SBDβ domain, demonstrating selective in silico affinity for tumor-expressed GRP78 [51, 52]. Also, there are several other ways to maintain this selectivity, such as targeting extracellular GRP78 with peptides like Pep42 or monoclonal antibodies avoid off-target effects on ER-localized GRP78 in healthy cells [53]. Additionally, GRP78-targeting nanoparticles offer tumor-specific drug delivery [54], while GRP78 expression profiling via immunohistochemistry or imaging supports patient stratification [55]. Combining these approaches with phytochemicals and conventional chemotherapies further enhances tumor-specific ER stress and apoptotic responses [56].”

Response to Reviewer #1

Comment 1: Abstract: This part is well furnished that details the outcome and findings. Yet the preceding part seems to be vague and better to keep it crispier with 2 lines.

Response: We thank the reviewer for their time and effort in reviewing our manuscript. In accordance with the reviewer’s suggestion, we have revised the introductory part of the Abstract section. We believe that this modification enhances the clarity and consistency of the manuscript. The updated lines are as follows:

Line 25-28: “Cancer remains a leading cause of mortality worldwide, with conventional therapies showing limited efficacy and high toxicity. The increasing incidence and therapeutic resistance necessitate alternative strategies. In this regard, phytochemicals have emerged as potential sources of developing safer and novel anti-cancer agents.”

Comment 2: Introduction: Seems to be very clear about the work plan. The author should concentrate on pinpointing the importance of the study rather than explaining its importance within a vast forum, especially in the first paragraph.

Response: Thank you for this valuable suggestion. In the revised manuscript, we have updated the Introduction section as recommended, specifically the first paragraph, placing greater emphasis on the significance and relevance of the present study. The revised lines are as follows:

Line 78-87: Conventional cancer therapies, including chemotherapy, radiotherapy, and immunotherapy, are often limited by multidrug resistance, lack of target specificity, and systemic toxicity [2], necessitating the exploration of alternative therapeutic strategies. Phytochemicals, a diverse class of bioactive plant-derived compounds, have demonstrated the ability to modulate critical upstream and downstream oncogenic processes such as oxidative stress, chronic inflammation, dysregulated signaling pathways, and expression of pro-tumorigenic proteins, while exhibiting low toxicity profiles. Therefore, elucidating the mechanistic insights by which selected phytochemicals exert anti-cancer effects is essential for validating their therapeutic potential and facilitating their development as integrative agents in evidence-based cancer treatment strategies [3].

Comment 3: A complete analysis of the isolated constituents has been well explicated with detailed analysis. A distinct analysis with appropriate parameters highlights the piece of work. All the figures and tables pertaining to the methodologies are appreciable, which supports the reader’s understanding.

Response: We sincerely thank the reviewer for their encouraging remarks and appreciation of our work. We are pleased to know that the detailed analysis of the isolated constituents and the clarity of our methodological presentation, including the supporting figures and tables, were found to be well-structured and informative. Such positive feedback reinforces the scientific value of our study and encourages us in our future research endeavors.

---

## [Editor Report · Decision Letter 1]

Targeted modulation of MMP9 and GRP78 via molecular interaction and in silico profiling of Curcuma caesia rhizome metabolites: A computational drug discovery approach for cancer therapy

PONE-D-25-13435R1

Dear Dr. Bhattacharya,

We’re pleased to inform you that your manuscript has been judged scientifically suitable for publication and will be formally accepted for publication once it meets all outstanding technical requirements.

Kind regards,

Prakash Palaniswamy, Ph.D

Academic Editor

PLOS ONE
---

## [Editor Report · Acceptance letter]

PONE-D-25-13435R1

PLOS ONE

Dear Dr. Bhattacharya,

I'm pleased to inform you that your manuscript has been deemed suitable for publication in PLOS ONE. Congratulations! Your manuscript is now being handed over to our production team.

Kind regards,

on behalf of

Dr. Prakash Palaniswamy

Academic Editor

PLOS ONE